# Aggregation state of *Mycobacterium tuberculosis* impacts host immunity and augments pulmonary disease pathology

Afsal Kolloli[1], Ranjeet Kumar[1], Pooja Singh[1,2], Anshika Narang[1], Gilla Kaplan[3], Alex Sigal[4,5,6] & Selvakumar Subbian [1✉]

In vitro phagocytosis of *Mycobacterium tuberculosis* (Mtb) aggregates (Mtb-AG), rather than similar numbers of single bacilli (Mtb-SC), induces host macrophage death and favors bacterial growth. Here, we examined whether aggregation contributes to enhanced Mtb pathogenicity in vivo in rabbit lungs. Rabbits were exposed to infectious aerosols containing mainly Mtb-AG or Mtb-SC. The lung bacterial load, systemic immune response, histology, and immune cell composition were investigated over time. Genome-wide transcriptome analysis, cellular and tissue-level assays, and immunofluorescent imaging were performed on lung tissue to define and compare immune activation and pathogenesis between Mtb-AG and Mtb-SC infection. Lung bacillary loads, disease scores, lesion size, and structure were significantly higher in Mtb-AG than Mtb-SC infected animals. Differences in immune cell distribution and activation were noted in the lungs of the two groups of infected animals. Consistently larger lung granulomas with large aggregates of Mtb, extensive necrotic foci, and elevated matrix metalloproteases expression were observed in Mtb-AG infected rabbits. Our findings suggest that bacillary aggregation increases Mtb fitness for improved growth and accelerates lung inflammation and infected host cell death, thereby exacerbating disease pathology in the lungs.

[1] The Public Health Research Institute at New Jersey Medical School, Rutgers University, Newark, NJ 07103, USA. [2] Department of Pulmonary, Allergy, and Critical Care Medicine, The University of Alabama at Birmingham, Birmingham AL35294, USA. [3] University of Cape Town, Cape Town 7925, South Africa. [4] Africa Health Research Institute, Durban 4013, South Africa. [5] School of Laboratory Medicine and Medical Sciences, University of KwaZulu-Natal, Durban, South Africa. [6] Max Planck Institute for Infection Biology, Berlin, Germany. ✉email: subbiase@njms.rutgers.edu

Mycobacterium tuberculosis (Mtb), the causative agent of tuberculosis (TB), has infected about a quarter of the global population[1]. Of the immunocompetent individuals exposed to Mtb, ~95% manifest immune-controlled latent infection while the remaining 5% progress to active TB[1]. The host and pathogen determinants that favor the progression of Mtb infection to active disease are complex and not fully understood. Immune suppression, as seen in HIV-infected individuals, is associated with loss of protective immunity and the development of full-blown TB[2]. In addition, exposure to a high dose of viable Mtb, exhaled or coughed by a patient with cavitary pulmonary disease, favors a more efficient spread of the infection and progression to active disease[3,4]. Moreover, specific properties of different clinical Mtb isolates can impact their ability to establish infection and progress to active disease, as seen in animal models and clinical studies[4–7].

Mtb has a natural tendency to form aggregates (AG) or clumps (a.k.a "cording") in broth cultures due to the presence of a waxy cell wall. Electron microscopy studies of pathogenic mycobacteria have shown cording in the lag, log, and stationary growth phases[8,9]. We recently reported that exposure of human monocyte-derived macrophages to Mtb-AG, compared with a similar number of single bacilli, interferes with the control of intracellular bacillary growth and results in efficient replication of the bacilli, in association with the death of the phagocyte[10]. Furthermore, in vitro efferocytosis of Mtb-loaded-dead macrophages resulted in more host cell death and increased bacterial proliferation[10]. It is important to note that Mtb-AG are often seen at the cavitary surface of pulmonary lesions and in the infectious aerosols exhaled by patients with active pulmonary TB[9,11,12]. Based on these observations, we hypothesize that host exposure to Mtb-AG, as opposed to single bacilli (Mtb-SC), may be an effective bacterial strategy for establishing an infection that is more likely to progress to active disease.

To test this hypothesis, we utilized a rabbit model of pulmonary Mtb infection, shown previously to develop disease pathogenesis similar to that seen in patients with TB. Animals were exposed to aerosolized Mtb-AG, and the outcome of infection was compared with animals infected with similar numbers of Mtb-SC. The kinetics of Mtb growth, disease pathology, and immune cell activation were monitored over time to determine whether Mtb-AG infection results in increased bacillary load and progression to more severe disease in the lungs.

## Results

### Infection with Mtb-AG results in consistently increased bacterial burden in rabbit lungs.

To determine the relative growth capacity of Mtb-AG versus Mtb-SC in vivo, rabbits were infected with aerosolized bacilli, and the lung bacillary load was evaluated over time. At $T = 0$ (3 h post infection), the number of bacterial colony-forming units (CFU) implanted was similar in both Mtb-AG and Mtb-SC infected rabbit lungs (Fig. 1a). A significantly higher increase in the bacillary load was noted in Mtb-AG-, compared wih Mtb-SC-infected lungs by 1 week post infection. The differential bacillary load between Mtb-AG and Mtb-SC was sustained at 2 and 4 weeks post infection, though it was only statistically significant at the later time point (Fig. 1a). The bacterial CFU data were consistent with and supported by the data from imaging fluorescent Mtb and bacterial chromosomal equivalent (CEQ) assays that measured the total bacillary load in the lungs regardless of their aggregation state (Fig. 1b and c and Supplementary Fig. 1a–d). A positive correlation was observed between the lung CFU and CEQ values ($r^2 = 0.7881$), as well as between CFU and Mtb fluorescence intensity($r^2 = 0.7837$) (Supplementary Fig. 1e, f). Together, these findings suggest that the

aggregation state enables early and efficient Mtb growth in the lungs of infected rabbits.

### Activation of the systemic immune response in Mtb-AG and Mtb-SC infected rabbits.

To establish whether Mtb-AG preferentially inhibited the optimal systemic acquired immune response following infection, we evaluated the effect of Mtb-AG and Mtb-SC infection on the host adaptive response by measuring the number of activated/proliferating CD4+ and CD8+ T cells in the spleen by flow cytometry. As expected, T cells from both Mtb-SC and Mtb-AG infected rabbits showed significantly higher proliferative response than the cells from uninfected controls at both 2 and 4 weeks post infection (Fig. 2a, b). However, the percent of activated/proliferating CD4+ and CD8+ T cells in response to antigenic stimulation was significantly higher in Mtb-AG than Mtb-SC-infected animals at both 2 and 4 weeks post infection. Together,

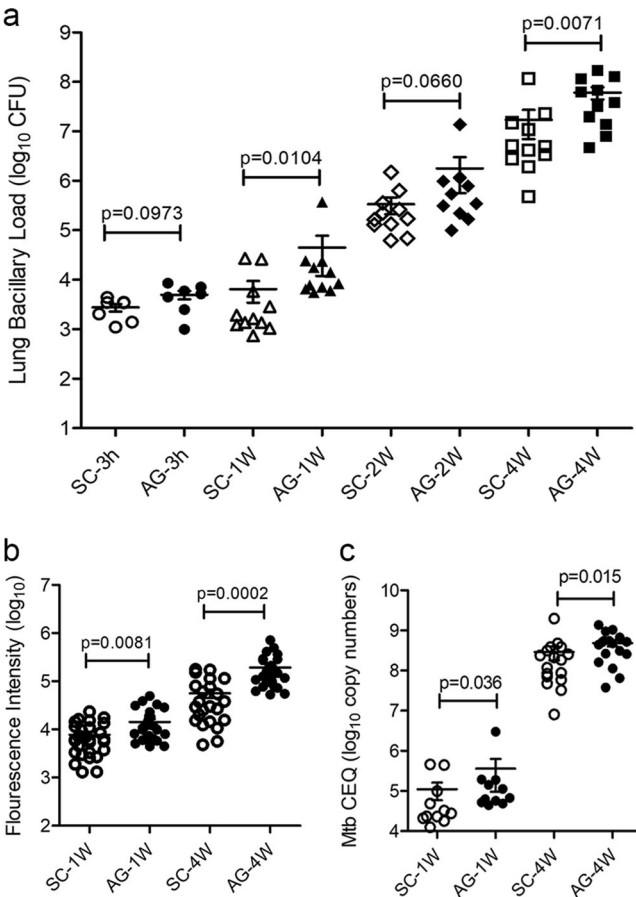

**Fig. 1 Growth kinetics of Mtb-AG and Mtb-SC in rabbit lungs. a** Bacillary load measured as the number of CFU in rabbit lungs infected with Mtb-AG compared with Mtb-SC at 3 h ($T = 0$), 1, 2, and 4 weeks post infection. Experiments were performed with $n = 6$–11 rabbits per group per time point. $p$ value was calculated using one-way ANOVA with Tukey's post-correction for multiple group comparison. **b** Net bacterial load in the lung sections as quantified by immunofluorescence of Mtb anti-LAM antibody at 1- and 4-week post infection. Data were compiled from measuring fluorescent intensity in 24–30 fields per lung ($n = 4$ rabbits per group per time point). $p$ value was calculated using Mann–Whitney $U$ Test . **c** Total lung bacterial load estimated by Mtb-CEQ analysis at 1- and 4-week post infection. Mtb copy numbers in lung samples were determined from a standard curve developed with a known concentration of Mtb genomic DNA isolated from broth cultures. Data were compiled from $n = 3$ rabbits per group per time point and repeated twice. $p$ value was calculated using Mann–Whitney $U$ Test.

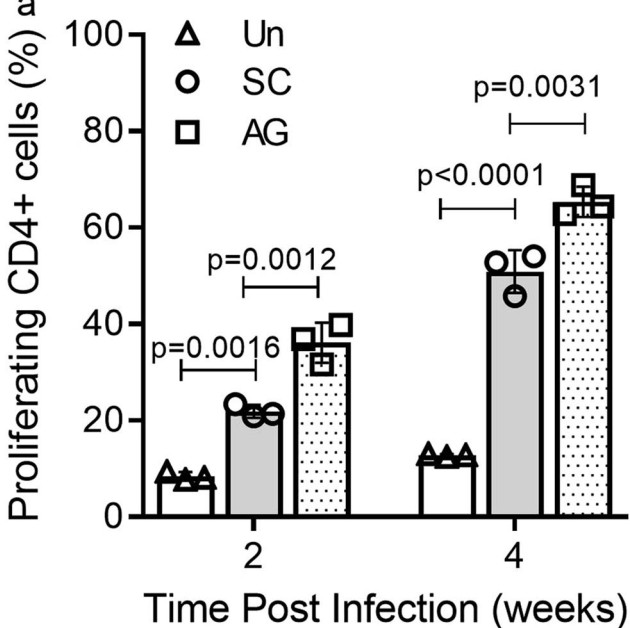

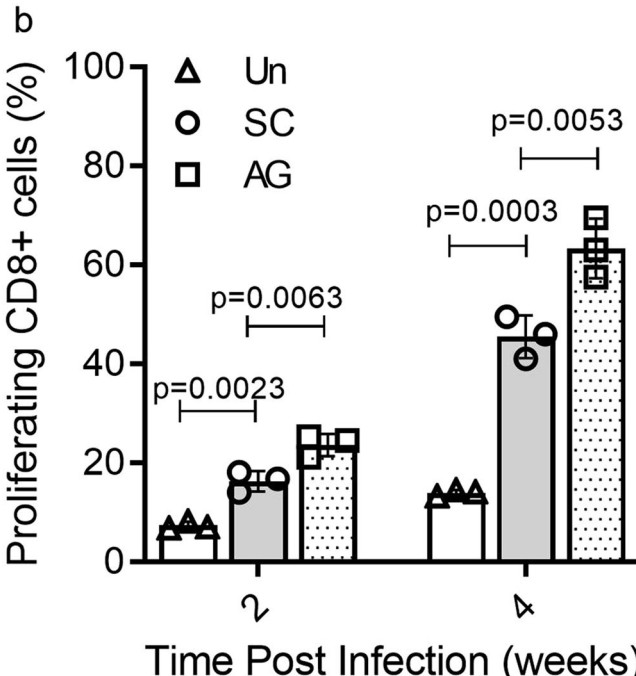

**Fig. 2 Activation of spleen CD4+ and CD8+ T cells in Mtb-AG or Mtb-SC infected rabbits.** The percentage of proliferating CD4+ (**a**) and CD8+ (**b**) T cells was determined at 2 and 4 weeks by flow cytometry analysis. Single-cell suspensions of the spleen were used and stained with cell-specific antibodies after in vitro stimulation with heat-killed H37Rv or unstimulated (Un). $n = 3$ rabbits per group. Data were analyzed by one-way ANOVA with Tukey's post correction for multi-group comparison.

these data suggest that a differential and robust immune activation occurs systemically as seen in the spleen, which is higher in response to Mtb-AG than Mtb-SC infection.

**Infection with Mtb-AG exacerbates disease pathology and affects immune cell distribution in the lungs.** We next evaluated the granulomatous response in Mtb-AG or Mtb-SC infected rabbit lungs (Fig. 3). The gross lung pathology at 4 weeks post infection showed smaller subpleural lesions in the Mtb-SC-infected group (Fig. 3a) than the Mtb-AG-infected animals (Fig. 3b). Histologic examination of H&E-stained lung sections showed multiple, coalescent lesions with several prominent necrotic foci with abundant polymorphonuclear cell accumulation, preferentially in the Mtb-AG-infected animals (Fig. 3d). In these lesions, Mtb bacilli were primarily seen as larger aggregates (Fig. 3f). In contrast, in the Mtb-SC-infected lungs, the lesions were less necrotic (Fig. 3c), and the bacilli were seen as singles or small aggregates (Fig. 3e). Although the difference in the number of subpleural granulomas was not statistically significant (Fig. 3g), the Mtb-AG-infected lungs had significantly larger lesions and a higher pulmonary disease score than Mtb-SC-infected animals (Fig. 3h, i). These observations suggest that Mtb-AG infection exacerbates pulmonary disease pathology, leading to more-extensive cell recruitment, inflammation, and necrosis than Mtb-SC infection.

The immune cell distribution in Mtb-AG or Mtb-SC-infected rabbit lungs were measured by flow cytometry of single-cell suspension (Fig. 4a, c) and immunohistochemistry (Fig. 4b, d; Supplementary Fig. 2). A similar number of viable CD11b+ innate immune cells, including monocytes, neutrophils, and natural killer (NK) cells, were observed in Mtb-AG and Mtb-SC-infected lungs at 2 weeks post infection. However, a significantly higher number of CD11b+ cells was noted in the Mtb-SC infected animals at 4 weeks. (Fig. 4a). In contrast, a significantly higher number of activated macrophages (IBA1+) was present in the lungs of Mtb-AG infected rabbits at 2 and 4 weeks post infection (Fig. 4b), which is consistent with the bigger lesion size in these animals. No significant difference in the number of CD4+ T cells was observed between the Mtb-AG and Mtb-SC-infected animals (Fig. 4c), whereas a significantly higher number of CD8+ T cells was noted in the Mtb-SC-infected rabbit lungs at 2 and 4 weeks post infection (Fig. 4d).

**Mtb-AG exposure upregulates necrosis and cell death networks early during lung infection.** RNAseq analysis was performed to evaluate the early (i.e., 24 h post infection) host response in rabbit lungs exposed to Mtb-AG and Mtb-SC. The RNAseq data of Mtb-infected groups were normalized to the corresponding data from uninfected control animals. The differential gene expression at the network/pathway levels was compared between Mtb-AG versus Mtb-SC groups. The RNAseq data obtained from all samples were consistent within replicate samples and reproducible across groups. The quality control, quality assurance, map reads, coverage, mapping alignment plot, and summary of gene counts of individual RNAseq data are shown in Supplementary Figs. 3–5. A higher number of genes were significantly perturbed in the Mtb-AG than Mtb-SC-infected rabbit lungs (1552 versus 1373 SDEG), with 822 genes commonly expressed between these two groups (Supplementary Fig. 6a, b). The pathway/network analysis of significantly differentially expressed genes (SDEG) revealed that the inflammatory response was upregulated to a greater extent in Mtb-AG-infected rabbit lungs (Table 1). The host cell death and organismal morbidity, and host cell necrosis were upregulated, whereas host cell survival and autophagy networks were downregulated during Mtb-AG, relative to Mtb-SC infection (Table 1; Supplementary Data 1–6).

Other significantly differently regulated networks included host cell viability, cell survival, molecular export, and cellular homeostasis (Table 1). Consistent with the perturbations in these biological functions, several canonical pathways associated with pro-inflammatory responses, including macropinocytosis signaling, granulocyte-macrophage colony-stimulating factor signaling, dendritic cell maturation, and Th1 responses, were significantly downregulated during Mtb-AG infection of the

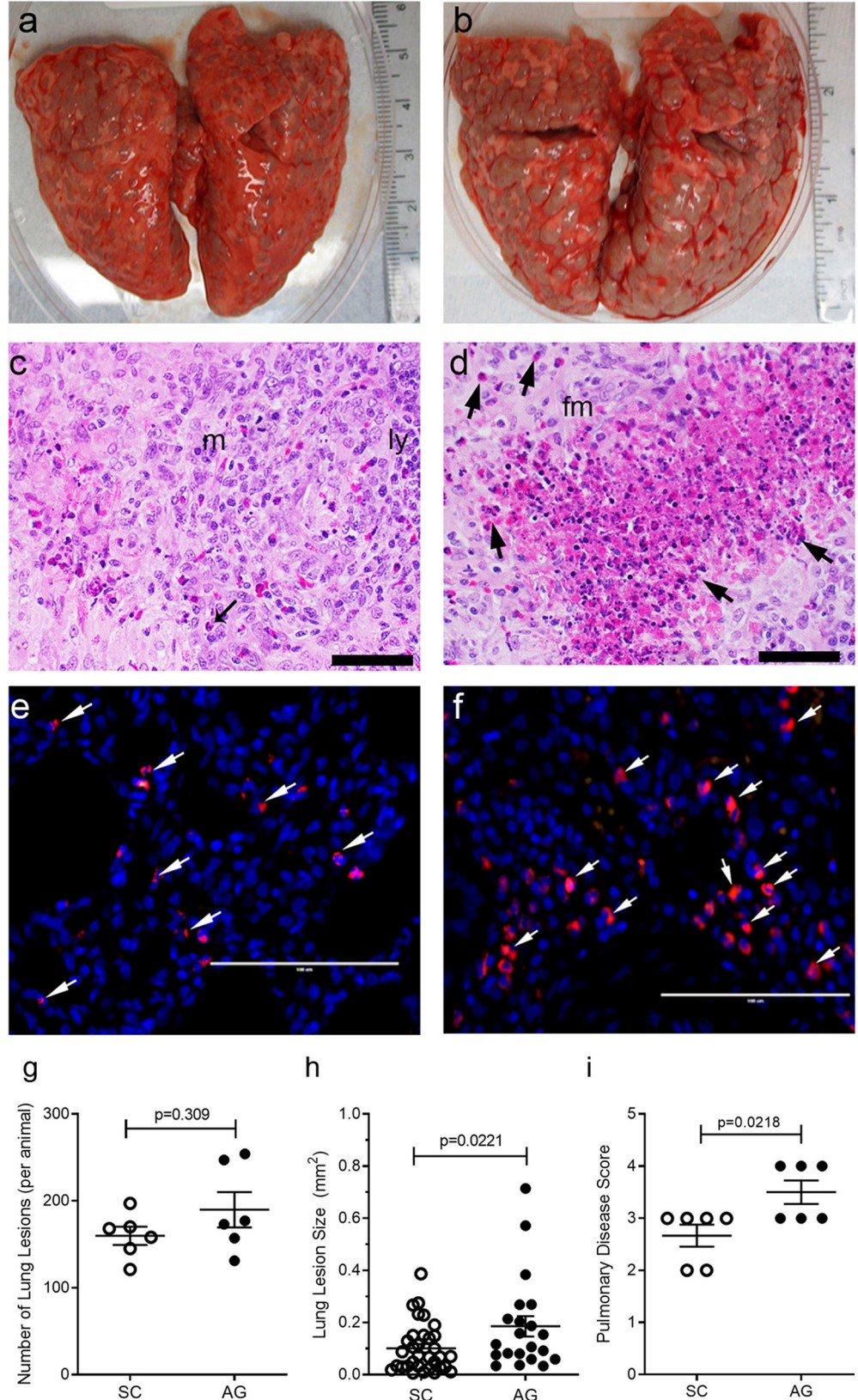

lungs. In contrast, genes involved in PTEN, PPAR, calcium, and CD27 signaling in immune cells were upregulated in Mtb-AG-infected lungs (Table 2). Thus, the RNAseq analysis revealed early (i.e., 24 h post infection) and more robust induction of host inflammatory responses and necrotic cell death pathways in

Mtb-AG infected rabbit lungs, compared with the Mtb-SC-infected animals.

**Mtb-AG induces early host inflammatory gene expression in the lungs.** Next, we investigated the expression pattern of early

**Fig. 3 Gross pathology and histopathologic analysis of Mtb-AG or Mtb-SC-infected rabbit lungs at 4 weeks post infection. a** Gross pathology of Mtb-SC infected rabbit lungs showing multiple, small, solid lesions. **b** Gross pathology of Mtb-AG infected rabbit lungs showing multiple, large, coalescent lesions. **c** Representative H&E-stained lung section of Mtb-SC infected rabbits showing a large granuloma with immune cell accumulation and small inflammatory foci surrounded by neutrophils (arrows), macrophages (m), and lymphocyte (ly) cuff. **d** Representative H&E-stained lung section of Mtb-AG-infected rabbits showing a granuloma with large necrotic foci surrounded by abundant neutrophils (arrows) and foamy macrophages (fm). **e, f** Immunofluorescence imaging of Mtb (arrows) in Mtb-SC (**e**) and Mtb-AG (**f**) infected rabbit lung sections. Original magnification: ×400 (**c, d**)×630 (**e, f**). Scale bar in **c–f** is 50 microns. **g** The number of subpleural lesions in rabbit lungs infected with Mtb-SC or Mtb-AG; *n* = 6 rabbits per group. *p* value was calculated using Mann–Whitney *U* Test. **h** Morphometric measurement of granulomatous lesion size in rabbit lungs infected with Mtb-SC or Mtb-AG. Data compiled from 24–30 fields per animal; *n* = 4 rabbits per group per time point. *p* value was calculated using Mann–Whitney *U* Test. **i** Pathological disease scoring of rabbit lungs infected with Mtb-SC or Mtb-AG. Data were obtained from *n* = 6 rabbits per group and analyzed by Mann–Whitney *U* Test.

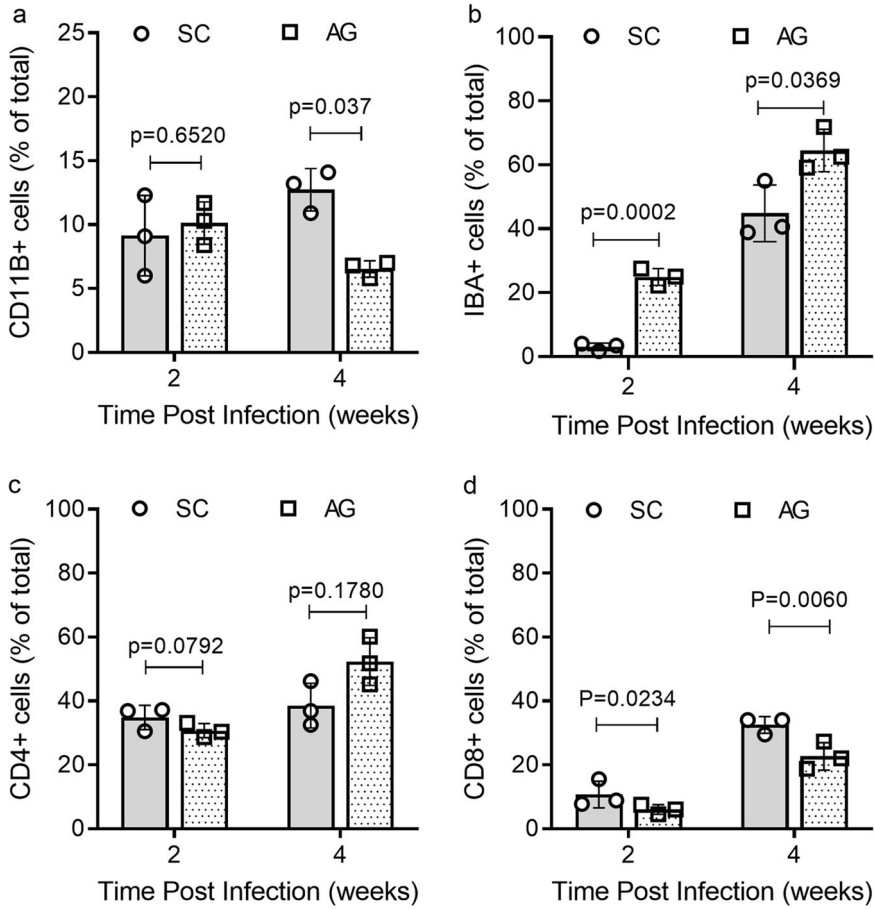

**Fig. 4 Immune cell distribution in the lungs after Mtb-AG or SC infection.** Lung cells (**a, c**) or tissue sections (**b, d**) from Mtb-SC or Mtb-AG infected rabbits at 2 and 4 weeks were stained with specific antibodies against. **a** CD11b, **b** IBA1 (macrophages), **c** CD4, and **d** CD8. Stained lung cells were analyzed by flow cytometry and stained lung sections were analyzed microscopically, and cells positive for the markers were counted manually in 40 random fields from each animal. *n* = 3 rabbits per group. Data were analyzed by Mann–Whitney *U* Test.

(i.e., 24 h post infection) inflammatory response markers in the lungs during Mtb-AG or Mtb-SC infection using quantitative real-time PCR analysis (qPCR). As shown in Fig. 5, infection with Mtb-AG significantly upregulated the expression of inflammatory molecules, such as *S100A8*, *S100A9*, *S100A12*, *CRP*, *HIF1A*, *MMP1*, *MMP9*, *TBX21*, and activated NK cell marker, *KLRG*, surface receptors, *PPARG*, *CD14*, as well as Th-2 type markers, *IL4* and *ARG1* (Fig. 5). In contrast, expression of antimicrobial molecules, including perforin (PRF1), cationic antimicrobial peptide (CAP18), and inflammasome (NLRP13), were upregulated in Mtb-SC-infected lungs at 24 h post infection (Fig. 5). The directionality of expression of several of these genes is consistent between the RNAseq and the qPCR data. These observations suggest that Mtb-AG induces a more robust host inflammatory response, whereas Mtb-SC upregulated antimicrobial host response in the infected rabbit lungs.

**Mtb-AG infection affects apoptosis, cytotoxicity, cell death, and dampens nitric oxide production in the infected lungs.** Since Mtb-AG triggers a more robust pulmonary inflammatory response and impacts activated immune cell distribution, we investigated the consequence of these processes on disease pathology by measuring the expression level of markers of host cell death in the rabbit lungs. Significantly elevated lactate dehydrogenase (LDH) levels (a measure of cell death) and host cell cytotoxicity were noted in the Mtb-AG-, compared with Mtb-SC-infected animals, as early as 1 week post infection and sustained until 4 weeks post infection (Fig. 6a, b). Similarly,

**Table 1 List of significantly affected biological networks in Mtb-AG infected rabbit lungs.**

| Biological functions annotation | Adjusted p value[a] | Activation z-score[b] | No. of total genes | % consistency and direction[c] |
|---|---|---|---|---|
| Cell viability | 0.0114 | −2.40 | 53 | 66 (downregulated) |
| Cell survival | 0.00453 | −2.31 | 55 | 66 (downregulated) |
| Molecular export | 0.0027 | −2.27 | 16 | 50 (downregulated) |
| Autophagy | 0.0128 | −2.26 | 17 | 82 (downregulated) |
| Cellular homeostasis | 0.0126 | −2.11 | 62 | 44 (downregulated) |
| DNA repair | 0.000409 | −2.0 | 19 | 17 (downregulated) |
| Inflammatory Response | 0.05 | 2.0 | 81 | 60 (upregulated) |
| Necrosis of cells | 0.00443 | 2.40 | 72 | 58 (upregulated) |
| Apoptosis | 0.00443 | 2.40 | 9 | 66 (upregulated) |
| Morbidity or mortality | 0.00465 | 4.36 | 92 | 64 (upregulated) |
| Cellular injury and organismal death | 0.00651 | 4.4 | 80 | 74 (upregulated) |

[a]Compared with Mtb-SC infection data.
[b]z-score is a measure of significance for the directionality of a biological function/pathway. It is derived from the gene expression pattern. A positive z-score indicates activation/upregulation, and a negative z-score indicates suppression/downregulation of a biological function.
[c]The percentage of total genes with expression consistent with the activation/upregulation or suppression/downregulation of a biological network.

**Table 2 List of up- and downregulated canonical host signaling pathways in Mtb-AG-infected rabbit lungs.**

| Canonical pathways | z-score (AG-vs-SC) and direction[a] |
|---|---|
| PTEN signaling | 2.79 (upregulated) |
| RhoGDI signaling | 2.65 (upregulated) |
| PPAR signaling | 2.42 (upregulated) |
| Calcium signaling | 2.40 (upregulated) |
| CD27 signaling in lymphocytes | 2.36 (upregulated) |
| GAS signaling | 1.70 (upregulated) |
| p38MAPK signaling | 1.52 (upregulated) |
| Macropinocytosis signaling | −1.89 (downregulated) |
| ERK/MAPK signaling | −2.06 (downregulated) |
| Dendritic cell maturation | −3.01 (downregulated) |
| VEGF signaling | −3.05 (downregulated) |
| FcgRIIB signaling in B cells | −3.72 (downregulated) |
| GM-CSF Signaling | −4.22 (downregulated) |
| Th1 Pathway | −5.24 (downregulated) |

[a]z-score is a measure of significance for the directionality of a biological function/pathway derived from the gene expression pattern. A positive z-score indicates activation/upregulation, and a negative z-score indicates suppression/downregulation of a biological function.

an increased host cell apoptosis was observed in the lungs of Mtb-AG-infected rabbits at 2 and 4 weeks post infection (Fig. 6d). In contrast, nitric oxide (NO) levels were significantly higher in Mtb-SC-infected, compared with Mtb-AG-infected, rabbit lungs at 1 week, although the difference was lost by 4 weeks post infection (Fig. 6c). These data suggest that Mtb-AG infection is associated with elevated cytotoxicity and the death of infected cells. Furthermore, the high production of NO, an antimicrobial molecule, during the early stages of infection might contribute to the control of Mtb-SC by the innate immune cells in the lungs. Consistent with this notion, Mtb-AG tolerated higher concentrations of agents that produce reactive oxygen species than Mtb-SC in broth cultures (Supplementary Fig. 7).

**Mtb-AG induces host inflammatory gene expression in the lungs.** We observed upregulation of inflammatory response genes in Mtb-AG-infected rabbit lungs as early as 24 h post infection, impacting the lung structure and function. Therefore, we investigated the inflammatory response genes that contribute to disease pathology by measuring the expression level of respective marker genes in the lungs of Mtb-AG or Mtb-SC infected rabbits. As shown in Table 3, Mtb-AG infection significantly upregulated the expression of inflammatory molecules S100A8, S100A9, CRP, IL23, activated NK cell marker (KLRG), and Th-2 type marker,

IL10 at 1 week post infection (Table 3). A similar gene expression pattern was noted in Mtb-AG-infected lungs at 4 weeks post infection with a significantly higher level of S100A8, S100A9, S100A12, IL17A, KLRG1, PPARG, ARG1, and IL10, and significantly dampened TBX21 expression (Table 3). At the protein level, Mtb-AG induced more MMP9, IL-17, and IL21, and lower TNF-α, IL-1α and NCAM levels than Mtb-SC at 4 weeks in the lungs, while IL-8, IL-1β, MIP-1β, and leptin levels were similar between these two groups (Supplementary Fig. 8). These observations suggest that induction of the host inflammatory response could exacerbate cell death, necrosis, and disease pathology in Mtb-AG-infected lungs.

**Mtb-AG exacerbates tissue remodeling in the lungs.** To determine the effect of Mtb-AG-induced inflammatory response on lung tissue remodeling, we investigated the expression of collagenases (MMP1, MMP13), stromelysin (MMP3), gelatinase (MMP9), elastase (MMP12), and membrane-type matrix metalloproteases (MMP14) by qPCR (Fig. 7a). A significant increase in the expression of MMP1, MMP3, MMP9, MMP12, MMP13, and MMP14, was noted in Mtb-AG infected, compared with Mtb-SC-infected rabbit lungs, at both 1 and 4 weeks post infection. Consistent with the upregulation of these MMPs, the Mtb-AG-infected rabbit lungs showed more fibrin and collagen deposition (Fig. 7b) than Mtb-SC-infected animals (Fig. 7c) at 4 weeks post infection, as revealed by the Masson's trichrome staining. These results suggest that active tissue damage/remodeling is associated with elevated MMP gene expression and inflammation mediated by necrotic host cell death in Mtb-AG-infected rabbit lungs.

## Discussion

Our recent observation has shown that in vitro infection of human monocyte-derived macrophages with Mtb-AG resulted in rapid phagocyte death, which was not seen with similar numbers of Mtb-SC[10]. In addition, in vitro efferocytosis of these cells by fresh macrophages caused rapid necrosis of the newly added/infected cells, leading to a cascade of host cell destruction[10]. Mtb-AG replicated efficiently in the infected/dead macrophages, whereas bacillary growth was limited in the Mtb-SC-infected macrophages. In the present study, we tested the hypothesis that a similar differential response would be seen in vivo. That is, exposure of the rabbit lung to Mtb-AG would confer superior survival fitness on the bacilli in association with more-extensive cell necrosis and lung pathology. Indeed, we observed that Mtb-AG infection resulted in a sustained higher lung bacillary load, which was not the result of a weakened systemic host immune

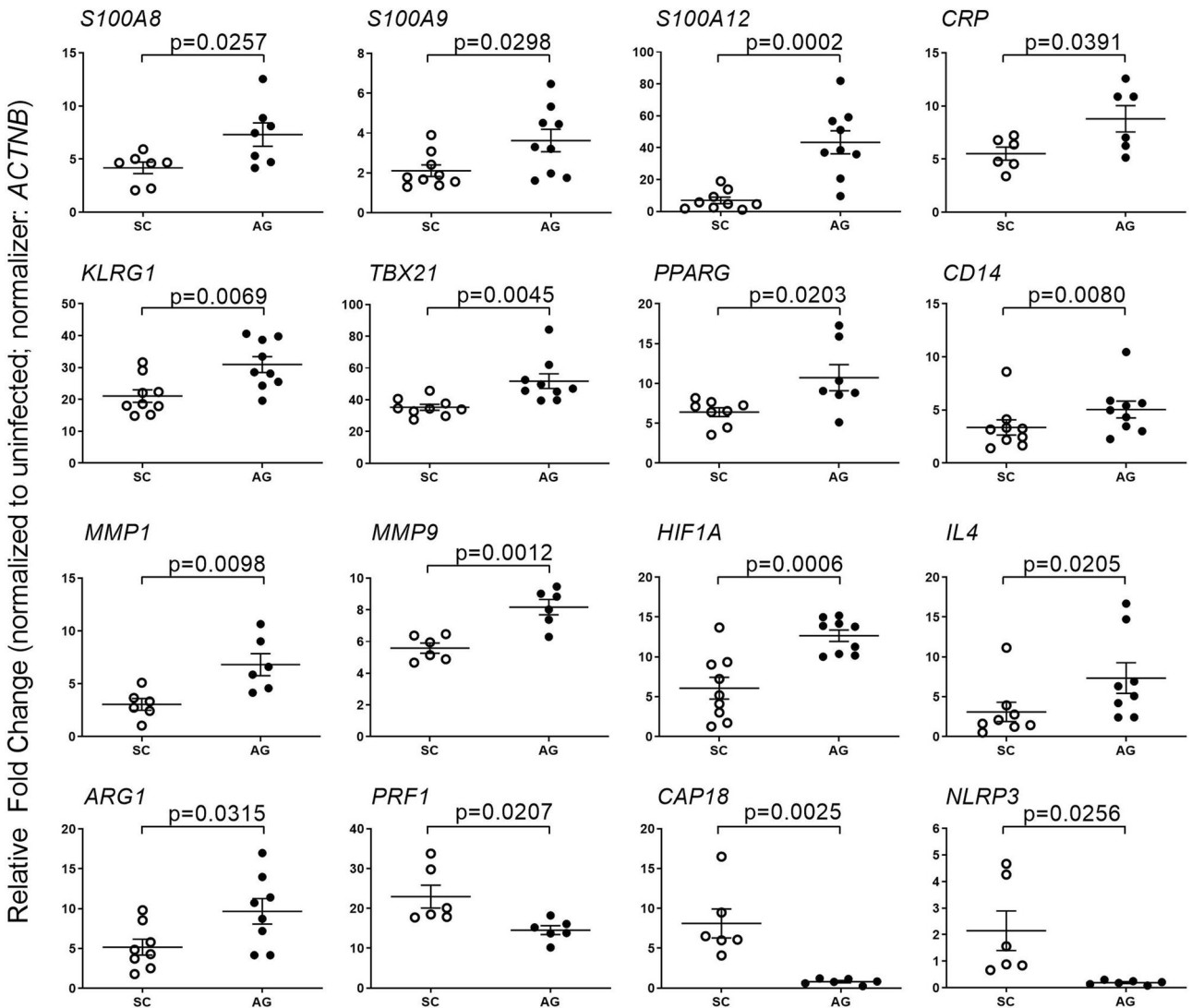

**Fig. 5 Modulation of host inflammatory response gene expression in Mtb-AG- and Mtb-SC-infected rabbit lungs.** qPCR was used to measure the expression of selected host inflammatory response genes at 24 h post infection. The expression level of genes in uninfected lungs was used to normalize their corresponding levels in Mtb-AG and Mtb-SC infected samples. Beta-actin gene (*ACTNB*) expression level was used as internal calibration control for each sample. The experiment was performed with $n = 3–4$ rabbits per group per time point and repeated at least twice. *p* value was calculated using Mann–Whitney *U* Test.

response. Rather, the effect appeared to be associated with the local, more severe pulmonary inflammation and extensive tissue remodeling seen in the lesions of Mtb-AG-infected animals. Consistently, several gene networks and pathways associated with necrotic cell death, cytotoxicity, and increased bacillary survival were more robustly upregulated early in Mtb-AG infected rabbit lungs. These observations suggest that the ability of lung phagocytes to restrict Mtb growth soon after phagocytosis would contribute to determining the subsequent extent of infection and disease pathogenesis.

Traditionally, both in vitro and in vivo experimental studies with Mtb have used deliberately dissociated aggregates/corded bacilli using mechanical (e.g., sonication) and chemical (e.g., Tween-80) methods. As single-cell suspensions generated by these methods facilitate an easier quantification of the bacilli used for infection and microbial characterization, the practice in the field has been to use Mtb grown, to the extent possible, as single cells. Consequently, the potential role of Mtb aggregation in driving the outcome of host–pathogen interactions and disease pathogenesis has essentially been overlooked and understudied.

Clearly, the number of bacteria in the infectious inoculum affects the outcome of the host response to Mtb, and a higher initial bacterial load leads to a more severe disease[7]. Similarly, the physiological form of Mtb (i.e, single cells versus aggregates) can impact the host–pathogen interactions and the outcome of infection in vivo. Clinical studies show that patients with active pulmonary TB produced aerosols with ≥10 Mtb CFU aggregates consistently had higher sputum AFB smear grades and transmitted more efficiently than those producing aerosols with >10 CFU[3].

Several bacterial factors could contribute to the differential outcome following infection of host cells by similar bacillary loads of Mtb-SC and Mtb-AG. Growing Mtb in media containing detergents, such as Tween-80, impairs the constituents of cell wall molecules, which negatively impacts bacterial virulence[13]. In contrast, Mtb grown without detergent in the media forms aggregates (a.k.a. cording phenotype) that are more resilient to killing by chemical agents and show increased virulence in human alveolar macrophages[9,14,15]. Furthermore, the cording of Mtb has been implicated in the induction of inflammation,

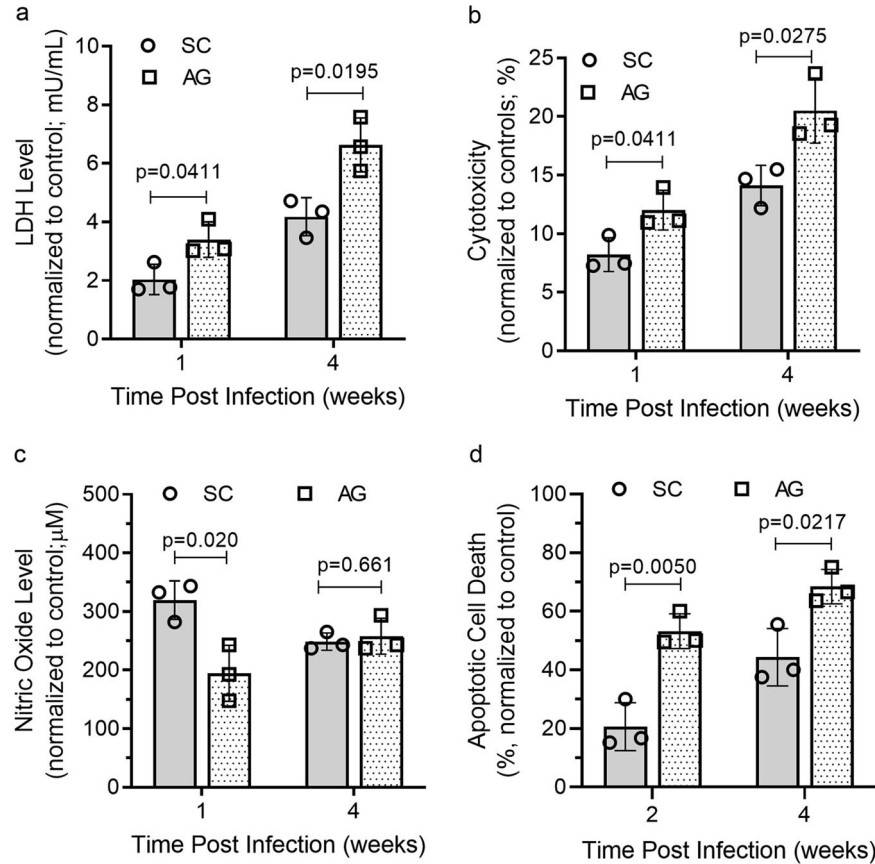

**Fig. 6 Mtb-AG exacerbates LDH levels, cytotoxicity, apoptosis, and dampens NO₂ levels in the infected rabbit lungs. a** LDH levels were measured colorimetrically in the lung homogenates and **b** cytotoxicity was calculated from the LDH levels. **c** Nitric oxide level was determined by the Griess method using filtered cell lysate from infected rabbit lungs. **d** Apoptotic cell death was calculated by TUNEL assay on rabbit lung sections at 2 and 4 weeks post infection. All parameters were compared against uninfected control animals at 1 or 2 and 4 weeks post infection. $n = 3$ rabbits per group. Data were analyzed by Mann–Whitney $U$ Test.

**Table 3 Expression level of host inflammatory response genes in rabbit lungs infected with Mtb-AG or Mtb-SC.**

| Gene | Fold change[a] (1-wk post infection) | | | Fold change[a] (4-wk post infection) | | |
|---|---|---|---|---|---|---|
| | SC infection | AG infection | *p* value | SC infection | AG infection | *p* value |
| *S100A8* | 1.23 ± 0.32 | 25.52 ± 7.5 | 0.0093 | 3.46 ± 0.14 | 4.61 ± 0.24 | 0.0023 |
| *S100A9* | 1.50 ± 0.36 | 4.93 ± 0.36 | <0.0001 | 3.01 ± 0.12 | 4.17 ± 0.24 | 0.0018 |
| *S10012* | 0.43 ± 0.05 | 3.20 ± 1.34 | 0.0672 | 1.19 ± 0.26 | 2.20 ± 0.25 | 0.0213 |
| *CRP* | 0.32 ± 0.08 | 86.6 ± 20.2 | 0.0016 | 7.36 ± 0.43 | 9.88 ± 1.33 | 0.104 |
| *IL17A* | 4.73 ± 0.34 | 5.93 ± 0.60 | 0.1177 | 5.67 ± 0.73 | 10.43 ± 0.99 | 0.0015 |
| *IL23* | 0.27 ± 0.07 | 9.63 ± 2.18 | 0.0016 | 2.82 ± 0.57 | 4.40 ± 0.88 | 0.1551 |
| *ARG1* | 4.56 ± 1.33 | 27.6 ± 13.6 | 0.1222 | 1.70 ± 0.17 | 2.20 ± 0.14 | 0.0484 |
| *IL10* | 1.22 ± 0.18 | 6.83 ± 1.84 | 0.0128 | 2.62 ± 0.24 | 4.32 ± 0.34 | 0.0025 |
| *KLRG* | 30.56 ± 2.90 | 65.76 ± 7.59 | 0.0015 | 8.17 ± 0.49 | 14.18 ± 1.55 | 0.0043 |
| *TBX21* | 3.808 ± 0.393 | 2.776 ± 0.37 | 0.0861 | 5.17 ± 0.18 | 4.13 ± 0.22 | 0.0048 |

[a]Relative to uninfected controls. GAPDH expression was used to normalize the expression of target genes in each sample. $n = 3$–4 per group per time point. The experiment was repeated twice, and data were presented as mean ± SEM.

impairing the acidification of phagosomes, subverting the bactericidal activities in infected macrophages, and promoting rapid disease progression in a murine model of TB[16–19]. It is important to note that Mtb isolated from the pulmonary cavities and exhaled air of TB patients have a cording phenotype[3,11,20]. Mtb aggregates contain elevated levels of trehalose 6,6' dimycolate, a major glycolipid present in the bacillary cell wall, which induces M1-type inflammation very early during Mtb infection of macrophages and contributes to necrotic granuloma formation in

infected mice[17,21]. These studies support our finding that Mtb-AG can cause severe inflammation and host cell destruction, resulting in larger, necrotic granulomas in infected rabbit lungs. Phagocytes, such as macrophages, dendritic cells (DC), and neutrophils, may utilize various phagocytic mechanisms to engulf Mtb[22,23]. For example, macrophages employ different modes of phagocytosis through Fc and C3 receptors, as reported earlier[23,24]. Importantly, the fate of intracellular Mtb is impacted by the type of phagocytic receptor involved in the bacterial

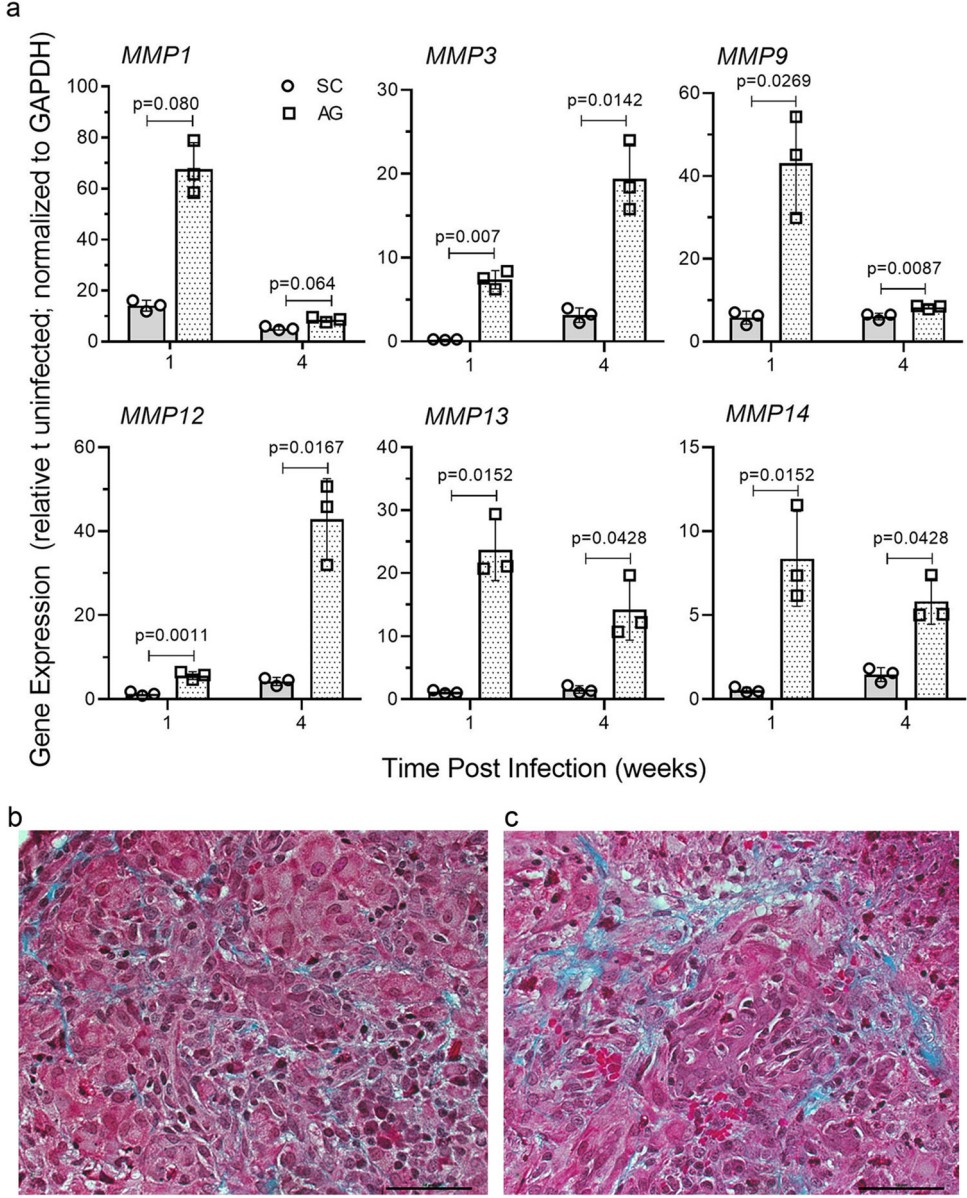

**Fig. 7 Mtb-AG upregulates MMP gene expression and induces more fibrosis in infected rabbit lungs. a** Expression level of MMP family genes was determined by qPCR using RNA from rabbit lungs infected with Mtb-AG or Mtb-SC at 1 and 4 weeks post infection. Expression levels of each target gene were normalized to GAPDH levels. Levels in uninfected animals were used to calibrate the values in Mtb-SC or Mtb-AG-infected groups. The experiment was repeated with three biological replicates in duplicates. Values plotted are mean ± standard error. Data were analyzed by Mann–Whitney $U$ Test. *$p < 0.05$, **$p < 0.01$. **b** Representative trichrome stained lung section of rabbits infected with Mtb-SC. **c** Representative trichrome stained lung section of rabbits infected with Mtb-AG showing the extent of collagen deposition and fibrosis (blue color) at 4 weeks. Original magnification: ×400 (**b**, **c**). Scale bar in **b** and **c** is 50 microns.

uptake[23,25]. However, the relative contribution of various phagocytes and their cell surface receptors involved in the uptake of Mtb-AG versus Mtb-SC is yet to be determined.

Pathogen-mediated host cell death has a crucial role in shaping the outcome of infection. Autophagy, a host cell homeostasis mechanism, has been implicated in controlling intracellular bacterial growth[26]. Virulent Mtb has been shown to downregulate autophagy and promote host cell lysis by pyroptosis, necroptosis, and necrosis, thereby evading the antimicrobial responses of the host phagocytes[27,28]. Moreover, as shown in this study, an early dysregulation of autophagy can trigger inflammation and contribute to elevated bacillary load and progression of disease pathology, as reported[29,30]. In addition, robust host cell necrosis combined with dampened autophagy leads to impaired tissue

remodeling and fibrosis[31]. In our studies, infection with Mtb-AG leads to the upregulation of the MMP family of genes and MMP9 protein, increasing tissue fibrosis in the rabbit lungs. Several studies, including ours, have shown the direct association between elevated MMP gene expression and remodeling of the lung extracellular matrix and increased tissue fibrosis[32–34]. Moreover, inhibition of MMP activity reduces morbidity and mortality of TB cases[35,36], and inflammatory cytokines such as IL-17 and IL-23 regulate MMP expression and activity[37]. In the present study, we show that Mtb-AG infection was associated with increased ARG1, IL-10, IL-17, and IL-23, potentially contributing to the induction of MMPs and fibrosis in the infected rabbit lungs.

In summary, we show that the nature of Mtb implanted in the lungs (i.e., Mtb-AG versus Mtb-SC) can impact the infection

outcome. We also demonstrate that the early host–pathogen interaction following uptake of Mtb as SC or AG, and the outcome of such interactions determine the rate and magnitude of disease progression in the lungs. Aerosols containing Mtb-AG, by inducing more host cell necrosis, contribute to rapid and robust disease progression, compared with Mtb-SC in rabbit lungs. Thus, the various clinical outcomes of Mtb infection in humans, i.e., latent infection, active disease, and bacterial transmission, may be determined by the very early host response to the mechanical properties of Mtb initially implanted in the lungs. Future studies are warranted to characterize the nature of Mtb in infectious air droplets and understand their physiology in the context of TB epidemiology.

## Methods

**Bacteria and chemicals.** Pathogenic *Mycobacterium tuberculosis* H37Rv (Mtb) strain was obtained from Dr. Shinnick (US-CDC, Atlanta, GA, USA). Mtb-mCherry strain was obtained from Dr. Sigal. Cultures of Mtb-AG and Mtb-SC were prepared by growing bacteria with or without Tween-80 as we described previously[10]. To determine the extent of aggregation, mCherry-expressing Mtb-AG cultures were processed to separate aggregates, and their fluorescent intensity was measured and normalized to the corresponding intensity from single bacteria. This analysis showed that the inoculum used for infection had $21 \pm 17$ bacilli per aggregate[10]. De-aggregation of broth-grown Mtb-AG into Mtb-SC was also achieved by ultrasonication to determine the number of bacteria as we reported previously[38]. All chemicals were purchased from Sigma (Sigma-Aldrich, St. Louis, MO, USA) unless mentioned otherwise.

**Rabbit aerosol infection.** Seventy-eight ($n = 78$) female New Zealand white rabbits (*Oryctolagus cuniculus*) of ~2.5 kg body weight were purchased from Covance Inc. (Covance Research Products, Denver, PA). Animals were randomly assigned into two groups and exposed to aerosols of Mtb H37Rv containing mostly Mtb-AG ($n = 39$) or Mtb-SC ($n = 39$) as described previously[38,39]. The rabbit aerosol infection chamber delivered bacterial inoculum of mostly Mtb-AG and Mtb-SC, as shown by staining of infectious aerosol particles collected at the delivery port directly on glass slides (Supplementary Fig. 9). The infectious inoculum of Mtb-AG and Mtb-SC was adjusted to a similar number of total live bacteria as SC, based on the CFU from de-aggregated Mtb-AG. A positive correlation was observed between the lung CFU and CEQ values, as well as between CFU and Mtb fluorescence intensity (Supplementary Fig. 1e, f).

At 3 h, 1 week, 2 weeks, and 4 weeks post infection, rabbits from each group ($n = 6$–11 per group per time point) were euthanized, and lungs and spleen were collected to evaluate bacterial CFU, histology, immune cell composition, enzyme levels, cellular function, and RNA isolation.

**Histopathology and morphometry.** Paraffin-embedded lung tissue was sectioned and stained with hematoxylin-eosin (H&E), acid-fast bacilli (AFB), or Masson's trichrome staining as previously reported[39]. Immunofluorescence with anti-Mtb primary antibody (Anti-Mtb-biotin antibody; 2 μg) (GeneTex, Irvine, CA, USA) and Texas Red-Streptavidin conjugated secondary antibody (1:1000 dilution) (ThermoFisher Scientific, Waltham, MA, USA) was used to visualize Mtb in lung sections as previously reported[40]. The stained sections were analyzed in a Nikon Microphot DXM 1200C microscope and photographed using NIS-Elements software (Nikon Instruments Inc., Melville, NY, USA). Mtb fluorescent intensity was quantified using Image J software (NIH, USA). Pathological analysis and disease scoring of lung sections was performed in a single-blinded way. The following scoring system was used: intact lung = 0, increased cellularity = 1, solid/cellular granuloma = 2, necrotic granuloma = 3, large coalescent necrotic granuloma = 4, partial liquefaction = 5, extensive liquefaction = 6 and cavitary lesion = 7. The number of subpleural lesions was counted manually on dissected lungs from each animal. The morphometric analysis to determine the size of lung lesions was performed using PathScan Enabler (Mayer Scientific, TX, USA), and the data were analyzed using SigmaScan Pro software (Systat Software, Inc, CA, USA).

**Immunophenotyping of lung cells.** Single-cell suspensions of rabbit lung cells were labeled with anti-rabbit-CD4-Alexa647 antibody (2 μg) (Novus Biologicals, Centennial, CO, USA), anti-rabbit-CD11b-PE antibody (2 μg) (Biorad, Hercules, CA, USA) to enumerate CD4+ T cells and innate immune cells, respectively, by using flow cytometry as described[38]. The gating strategy used for flow cytometry is shown in Supplementary Fig. 10. Immunofluorescent staining with anti-rabbit-CD8-Alexa Flour 488 antibody (2 μg) (Novus Biologicals, Centennial, CO, USA) and rabbit-cross reactive, anti-human IBA1 antibody (1:100 dilution) (Abcam, MA, USA) was performed to determine the number of CD8+ T cells and macrophages, respectively. Total and differential immune cells were counted in 20 fields per lung section per animal ($n = 3$ per group per time point) using a Zeiss 200 M fluorescent microscope (Carl Zeiss Microscopy LLC, White Plains, NY, USA).

**T-cell proliferation assay.** Splenocytes were isolated from Mtb-AG and Mtb-SC infected rabbits ($n = 3$ per group per time point) and labeled with carboxy-fluorescein succinimidyl ester (CFSE) as described previously[41]. Labeled cells were stimulated with heat-killed Mtb H37Rv or left unstimulated for 96 h, followed by staining with primary mouse anti-rabbit CD4 APC (1:100 dilution) (ThermoFisher Scientific, CA, USA) or anti-rabbit CD8 APC (ThermoFisher Scientific, CA, USA) antibodies and secondary goat anti-mouse IgG-APC antibody (ThermoFisher Scientific, CA, USA). Cell counts were measured and analyzed using BD Accuri C6 flow cytometer following the manufacturer's instructions (BD Biosciences, CA, USA).

**Cytokine measurement.** The levels of CCL4, CXCL8, MIP1B, IL1A, IL17A, IL1B, IL21, TNFA, MMP9, and NCAM-1 were measured in the rabbit lung homogenates ($n = 3$ per group per time point) using Quantibody Rabbit Cytokine Array following manufacturer's instructions (RayBiotech, Norcross, GA, USA).

**Quantification of NO.** Nitrite concentration in the lung homogenate was measured ($n = 3$ per group per time point) using the Griess Reagent kit following manufacturer instructions (Sigma, St. Louis, MO, USA). Absorbance was recorded using a GloMax plate reader (Promega Corporation, Madison, WI, USA).

**LDH activity, cytotoxicity, and apoptosis assays.** The LDH activity and percentage of cell cytotoxicity in rabbit lungs ($n = 3$ per group per time point) were determined using the LDH-Glow cytotoxicity kit as per the manufacturer's instructions (Promega Corporation, Madison, WI, USA). Luminescence was recorded using a GloMax plate reader. Host cell apoptosis was measured in rabbit lung sections using the DeadEnd Colorimetric terminal deoxynucleotidyl transferase dUTP nick end labeling System following the manufacturer's instructions (Promega Corporation, Madison, WI, USA). Apoptotic and non-apoptotic cells were counted manually using the EVOS FL microscope (ThermoFisher Scientific, CA, USA).

**Lung RNA isolation.** Total RNA from uninfected, Mtb-SC, and Mtb-AG-infected rabbit lungs ($n = 3$ per group per time point) was isolated using TRIzol reagent (Life Technologies, Grand Island, NY, USA) as described earlier[6,42]. The quality and quantity of the purified RNA were assessed by Bioanalyzer (Agilent Technologies Inc., Santa Clara, CA).

**RNAseq analysis.** Total lung RNA from Mtb-SC or Mtb-AG infected or uninfected (control) rabbit at 24 h post infection was used for RNAseq analysis with three animals per group ($n = 3$). RNAseq experiment was performed using Lexogen QuantSeq 3′mRNA-Seq Library Prep Kit FWD for Illumina and NextSeq sequencing using standard protocols developed and followed by Lexogen (Lexogen, Inc, Greenland, NH, USA). Briefly, QuantSeq FWD Kit (Lexogen, Inc, Greenland, NH, USA) was used for the mRNA-seq library preparation. To generate the library, total RNA was used for first-strand synthesis with oligo dT containing the Illumina-specific Read 2 linker sequence. After first-strand synthesis, the RNA was removed, and the second strand synthesis was initiated by random primer containing the Illumina-specific linker sequence. The second strand synthesis was followed by a magnetic bead-based purification step and library amplification step. A total of nine libraries (three biological replicates of Mtb-AG, Mtb-SC, and uninfected lung RNA) were single-end sequenced on an Illumina HiSeq 4000 instrument (Illumina, San Diego, CA, USA). Adapter sequences, poly-A, and low-quality reads were removed using TrimGalore software 16. Sequence alignment was performed with STAR aligner (Lexogen, Inc, Greenland, NH, USA) on the rabbit (*Oryctolagus cuniculus*) reference genome from NCBI (Taxonomy ID:9986). A total of $2.6 \times 10^8$ reads with an average of 71.37% uniquely mapped reads were obtained from nine samples. SDEG were selected based on a $p < 0.05$ and two-fold change cutoff in individual gene expression and used as input for network/pathway analysis using Ingenuity Pathway Analysis (IPA; Qiagen, Valencia, CA, USA) as we described previously[6]. In brief, the IPA knowledgebase for gene functions was used to derive differentially affected biological functions, networks, and canonical pathways in the RNAseq dataset. A z-score of $>+2$ (upregulated function) or $-2$ (downregulated function) was used to determine statistical significance between groups. The RNAseq metadata files have been submitted to Gene Expression Omnibus (GEO; Submission# GSE176139).

**Quantitative real-time PCR analysis.** Total RNA isolated from Mtb-AG and Mtb-SC infected rabbit lungs ($n = 3$–4 per group per time point) was used in qPCR experiments to measure the expression levels of *ARG1, TNFA, IL10, IL4, PPARG, S100A8, S100A9, S100A12, CRP, CD14, CD11B, NLRP3, CAP18, KLRG1, TBX21, HIF1A, ARG1, PRF, MMP1, MMP3, MMP9, MMP13, MMP14, IL10, IL17A* and *IL23* using Affinity QPCR cDNA Synthesis kit and Brilliant III Ultra-Fast SYBR® Green QPCR Master Mix as per kit instructions (Agilent Technologies Inc., Santa Clara, CA, USA). Target gene expression was normalized to beta-actin (ACTB) or GAPDH expression level in the same sample. The expression level of Mtb-AG and Mtb-SC were normalized to corresponding levels in the uninfected control samples.

The nucleotide sequence of primers used to amplify target genes in the qPCR experiment is listed in Supplementary Table 1.

**CEQ assay**. Mtb genomic equivalent, a measure of total bacterial load in the rabbit lung ($n = 3$ per group per time point), was determined by qPCR using Mtb16s rRNA gene and Brilliant III Ultra-Fast SYBR® Green QPCR Master Mix (Agilent Technologies, Inc. Santa Clara, CA, USA) as described earlier[43]. A positive correlation was observed between the lung CFU and CEQ values ($r^2 = 0.7881$) (Supplementary Fig. 1e).

**Mtb susceptibility test**. Mtb-AG and Mtb-SC were grown in 7H9 media with different concentrations of hydrogen peroxide. Bacterial viability was determined by Alamar blue assay as described previously[44].

**Statistics and reproducibility**. Mann–Whitney $U$ Test and one-way analysis of variance (ANOVA) with Tukey's correction were applied for pairwise and multiple group comparisons, respectively, using Prism-5 (GraphPad Software, La Jolla, CA, USA). A $p < 0.05$ was considered statistically significant. $*p < 0.05$, $**p < 0.01$, $***p < 0.005$. The sample size was calculated based on similar studies published in the literature. The number of biological samples used for each experiment is mentioned in the respective figure legends and methods section. Rabbits were randomly allocated into different infection groups and for different experimental time points.

**Study approval**. The Institutional Biosafety Committee and the Institutional Animal Care and Use Committee (IACUC) of Rutgers University approved all procedures involving rabbits. Animals were handled humanely according to the ethical guidelines of the federal Animal Welfare Act (AWA) and the Association for Assessment and Accreditation of Laboratory Animal Care International (AAALAC).

**Reporting summary**. Further information on research design is available in the Nature Research Reporting Summary linked to this article.

## Data availability

The data sets generated during and/or analyzed during the current study are available as supplementary data files linked to this manuscript. The RNAseq data is available at the Gene Expression Omnibus portal (Accession number: GSE176139). All other data are available from the corresponding author on reasonable request. Source data are deposited in Figshare[45].

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

## Acknowledgements

S.S. and A.S. gratefully acknowledge the Bill and Melinda Gates Foundation for funding this study (# OPP1116944).

## Author contributions

S.S. and A.S. conceived the concept. S.S., A.S. G.K. designed the research studies, A.K., R.K., P.S., A.N., and S.S. conducted experiments, acquired and analyzed data, A.K., S.S., and G.K. wrote the manuscript. All authors have read, edited, and agreed to the submission.

## Competing interests

The authors declare no competing interests.
