## [Transparent Peer Review File · Communications Biology]

Reviewers' comments:

Reviewer #1 (Remarks to the Author):

The MS don't contain any page number or line number, without page number and line number it is very difficult to guide authors to the extract query that are raised.

The manuscript (MS) "Aggregation state of Mycobacterium tuberculosis impacts host immunity and augments pulmonary disease pathology" by Kolloli et., al shows that aggregated mycobacteria causes greater pathogenicity and severe diseased condition in rabbit model owing to their better growth and greater survivability in comparison to similar number of single Mtb. They have also shown early induction of proinflammatory markers, prompt onset of cell death, increase in lung innate immune cell number, bigger lesion size in aggregated mycobacteria infected rabbits compared to rabbits infected with similar number of single Mtb.

General observations

- 1) The main theme of this MS is aggregation of mycobacteria, but how is the extent of aggregation determined, produced or maintained is not well described. When would a clump of M.tb considered as M.tb-AG should be substantiated. Is Mtb-Ag a range starting from two single M.tb to n numbers, then this range should be described and the reason for considering this range explained. There is no in vitro or in vivo control on the extent of aggregation during intracellular proliferation of M.tb. How are we to expect that M.tb-AG with n number of M.tb when proliferate would further generate more Mtb-AG and not few Mtb-SC as well. If the later case occurs then that would give mixed results. Similarly when infection is carried out with few Mtb-SC's there is no control that these M.tb-SC would not aggregate together and form M.tb-AG and thus give mixed results. Nothing has been mentioned what measures if any has been taken by authors to maintain Mtb-SC as single cells all through.
- 2) Heavier aerosolic droplets tend to get retarded in the pharynx and bronchi based on their size. Based on equal CFU values of Mtb-AG and M.tb-SC at time 0, the authors conclude that equal amount of M.tb was presented to the lungs for infection. If this is the case then compared to a single phagocytosis event for an n no. of Mtb containing Mtb-AG, n no of phagocytic events have to occur for M.tb-SC. This is possible only if phagocytosis is considerably slower for M.tb-AG than for M.tb-SC and if that is the case then how can pathogenesis of two differentially phagocytosed populations be compared.
- 3) A more detail description of RNAseq and network analysis done for SDEG will be helpful
- 4) As gene expression pattern in Mtb-AG infected rabbit lungs revealed induction of both apoptosis and necrosis, so it will be better to make a percentage wise comparison between apoptosis and necrosis mediated cell death and their outcome on granuloma formation .
- 5) As immune cell composition has been investigated, to ascertain proper spatial localization along with increase in number of different innate immune cell types, multiparameter immunofluorescence imaging of M.tb-AG and M.tb-SC infected lung tissue sections exhibiting CD11b, CD4, F4/80, B220, M.tb, SIGLEC antibody should be carried out. This data will help in better comparison of infection exacerbated pulmonary disease pathological condition between Mtb-AG and Mtb-SC infected rabbits .
- 6) For better analysing spatial location of different innate immune cells machine learning clustering algorithms should be used.
- 7) A complete list of used antibodies with species specificity is required in the materials and methods section. A comprehensive list of all the oligonucleotide sequences used in this study should be provided.
- 8) More discussion on bigger lesion size and greater induction of necrosis in Mtb-AG infected rabbits is required corresponding to macrophage, T-cell, neutrophil spatial location and their distance in between should be provided.
- 9) In majority of cases independent of host status differential killing of bacteria take place and majority of the granulomas are established by a single bacterium then why is lesion size of Mtb-AG infected rabbits bigger than lesion size of rabbits infected with Mtb-SC. A quantitative estimation of the lesion size and the mechanism used to measure the same should be provided.

10) Tuberculosis spread through aerosolic droplet nuclei measuring 1-5 μm and these droplets contain 1-4 bacteria, which then reach the alveoli, are engulfed by macrophages and thereafter establish an infection leading to granuloma formation as has been shown in case of Mtb-SC. While aerosols $> 7\mu\text{m}$ tend to get retarded at the pharyngeal region which if successful in establishing an infection would in a non-macrophage cell type or in extra-pulmonary settings. Therefore what is the physiological significance of a forced non-physiological Mtb-AG infected rabbit model, where M.tb-AG containing at least 20 M.tb forced into aerosol formation for infection. Such M.tb-AG under normal condition cannot be spread as aerosol droplet from one person to another.

11) How is the M.tb-SC maintained as single cells once it enters inside the respiratory track of the rabbit?

12) Role of TNF alpha cannot be understated in case of initial disease spread. Why TNF alpha has been measured after 4 wk of infection and not after 1 wk of infection when bacteria starts to get disseminated.

Specific points

1. Introduction-last para-3rd line-'aerosolized Mtb-AG': If Mtb-SC is single M.tb then what is M.tb-AG two bacteria onwards. This is unclear. Proper experiments should be included which makes a clear distinction when would a M.tb clump be designated as M.tb-AG.

2. Introduction-last para-4th line-'similar number of M.tb-SC: what is meant by similar number of M.tb-SC. If M.tb-AG constitutes a specific number of M.tb forming an aggregate then this can be considered, but if M.tb-AG is a range of M.tb aggregates then how are the authors monitoring the similar number of M.tb-SC each time. On the contrary if M.tb-AG is a fixed number of M.tb containing aggregate then how do the authors obtain this specific aggregate should be stated explicitly. If M.tb-SC contains same number of M.tb singly that are present in M.tb-AG then how do the authors make sure that these M.tb-SC's do not form mini M.tb-AG's within themselves. Besides how do the authors also control the formation of macro M.tb-AG by coming together of 2 or more M.tb-AG.

3. Results-Section 1-3rd line- if the CFU is similar then the no. of M.tb whether it be M.tb-AG or M.tb-SC are similar. Arbitrarily considering that M.tb-AG is constituted by 5 M.tb, then CFU would be similar if the extent of phagocytosis in M.tb-AG is 5 fold less than M.tb-SC. If the phagocytosis is similar then the CFU would be 25 times than that of M.tb-SC. Besides to obtain CFU, M.tb from within the macrophage is obtained by lysing and then plated. So the CFU would be similar only when the extent of phagocytosis of M.tb-AG would be that many fold less as to the number of M.tb present in the M.tb-AG. If the M.tb-AG is a range then it would be more complex proposition. If one colony is formed by each M.tb-AG then the colony size would vary according to the number of M.tb present in each M.tb-AG. In such a case it would not be a true CFU.

4. Results-Section 1-4th line- The significant increase in cfu could be attributed to lower doubling time of Mtb-AG as compared to M.tb-SC as well as comparable number of M.tb being present inside macrophages at 0 timepoint. If the latter is true then phagocytosis of M.tb-AG is much less than M.tb-SC at a given time. Therefore, raising the question of feasibility of comparing intracellular proliferation when the extent of uptake is different.

5. Results-Section 1-8th line-'imaging fluorescent'-representative images should be provided in the supplementary or main text.

6. Results-Section 1-8th line-'net bacillary load'-how do the authors ensure that the M.tb-AG is taken up by macrophages and not by neutrophils. In one of their previous studies it was shown that M.tb-AG is taken up by cells isolated from donor blood using CD14+ MACS beads which they termed MOM. It is to be noted that neutrophils do express CD14 to varying extents and would thus be isolated along with MOM.

7. Results-Section 2-5th line-'polymorphonuclear cell accumulation'-using a distinct set of markers the authors need to show conclusively that the uptake of non-physiological Mtb-AG occurs by macrophages and not by any other cells type such as neutrophils.

8. Results-Section 2-8th line-'small clumps'- why would these clumps not be designated under M.tb-AG and thereby whatever pathophysiology has been attributed to M.tb-AG from initial stage should be attributed to M.tb-SC from the second granuloma onwards.

9. Results-Section 2-10th line-'subpleural...Mtb-SC'-in the corresponding figures the authors have only

provided H-E staining and provided pointers for macrophages and neutrophils. The authors need to provide conclusive immunohistochemistry and/or immunofluorescence data exhibiting the macrophage to neutrophil distribution and the extent of infection therein for such sections from Mtb-AG and M.tb-SC infected lung sections.

10. Results-Section 2-13th line-'more....infection'- M.tb-AG exhibits more extensive cell recruitment as compared to Mtb-SC. M.tb infection is a form of stress. Highly stressed neutrophils are known to secrete myeloperoxidases (MPO) that then can induce severe inflammation and extensive cell recruitment. Authors should provide experiments to negate such a proposition.

11. Results-Section 2-16th line-'flow cytometry....immunohisto'-the authors used single cell suspensions to analyse the different cell types in the infected lungs. A flow cytometry data corresponding to stained M.tb along with other cell type is required to establish the context of infection here.

12. Results-Section 2-20th line-'significantly....four weeks'-naive neutrophils are also known to express higher levels of CD11b compared to the activated/stressed ones. Hence in 4weeks for M.tb-SC when true granuloma is formed lots of naive neutrophils are recruited which may not be the case for M.tb-AG which already could be inside a activated/stressed neutrophil and therefore does not induce further recruitment of naive neutrophils. Such propositions drawn from the provided data should be experimentally negated.

13. Results-Section 3-8th line-'inflammatory....upregulated'-since RNA seq analysis was carried out from different cell types in the lung tissue it would be hard to envisage whether the enumerated expression changes in Mtb-AG infected lungs as compared to Mtb-SC infected lungs is from the same cell type or not. So experiments exhibiting that for the presented data, both M.tb-AG and M.tb-SC is taken up by same cell type should be included.

14. Results-Section 3-12th line-'induction....necrosis'-In context of Mtb-AG infection only the infected cells, die at the initial stage. So how is it that the same cell is dying by necrosis as well as apoptosis simultaneously.

15. Results-Section 3-17th line-'macropinocytosis signalling'-From figure 1 it is evident that M.tb-AG is taken up through reduced phagocytosis. This could be circumvented if it is to be considered that activated macrophages which are Iba+ engulf these M.tb-AG. It is known that activated macrophages/monocytes uptake mycobacteria through macropinocytosis as the receptor mediated uptake for these clumps may not be possible and it also allows for uptake to a large extent. So it is surprising that genes corresponding to macropinocytosis is downregulated in M.tb-AG infected lungs.

16. Results-Section 3-19th line-'PPAR'-how do the authors specifically say that this upregulation of PPAR pathway genes are from Mtb-AG infected macrophage/monocytes and not from other cell types such as neutrophils as similar upregulation is observed there as well when they are dying by apoptosis.

17. Results-Section 4-3rd line-'upregulated.... inflammatory'-through the data presented in Table 2 the authors state that genes responsible for Th1 response are downregulated in Mtb-AG infected lungs. It is known that Th1 response corresponds to a pro-inflammatory response. But here the authors now show that genes corresponding to inflammatory response is upregulated in the lungs upon Mtb-AG infection. What would be the explanation to this contradiction.

18. Results-Section 4-9th line-'while....lungs'- this is strong statement to make based on just few gene upregulation data that too from the gross lungs and not the niche cells. This statement should be altered.

19. Results-Section 5-4th line-'percentage...Mtb-AG'-Previously the authors state that genes corresponding to both apoptosis and necrosis were upregulated in the lungs upon infection with M.tb-AG. But here they exhibit only biochemical markers corresponding to apoptosis, this is surprising. Besides both Mtb-AG and Mtb-SC have been shown to form granulomas wherein granulomas are known to have a necrotising centre. Through these data do the authors suggest that granulomas of M.tb-AG have a apoptotic and dead cell core instead of a necrotic core which they had suggested earlier.

20. Results-Section 5-7th line-'nitric oxide...infection'-upon phagocytosis of a pathogen the prime macrophage response is to kill the bug by barging it with ROS and RNI. Overproduction of ROS and RNI in a cell can lead to apoptosis of the cell. The precursor for RNI is NO. Considering that upon Mtb-

AG infection the infected cells are the ones that die due to apoptosis it is quite surprising to find that these cells although dying by apoptosis exhibits reduced levels of NO. Additionally authors in earlier figure have shown Iba1+ cells in the lungs of M.tb-AG infected cells which is indicative of activated macrophages and monocytes. It is known that such activated macrophages and monocytes have higher RNI and ROS.

21. Results-Section 6-21th line-'four weeks'-Granuloma formation occurs both in Mtb-AG and Mtb-SC infected lungs. Overexpression of MMP's are associated with granulomas. Therefore the higher levels of MMP expression in Mtb-AG infected lungs 4weeks post infection is interesting.

Reviewer #2 (Remarks to the Author):

In their manuscript entitled "Aggregation state of Mycobacterium tuberculosis impacts host immunity and augments pulmonary disease pathology", Kolloli et al, describe the important role of aggregation in M. tuberculosis virulence. The authors use multiple approaches to show that bacterial aggregation causes more lung inflammation and damages in rabbit lungs. The manuscript is very well written and easy to follow. Here are my commentaries to the authors:

1/ As one clump gives one CFU, more bacteria could have been present in the Mtb-AG in comparison to Mtb-SC and being responsible to the phenotypes observed. But the authors have made the effort to show the good correlation between CFUs, CEQ and fluorescence to convince us that the number of bacteria compared is the same.

2/ Figure 4. The figure and figure legends should be modified to improve clarity. It is unclear what do "increased" and "decreased" values mean and why those categories show both negative and positive values. Also, it is not clear to me if the data presented are fold changes or z-scores. In the legend, reference is made to Figure 4D but there is not figure 4D. The addition of gene names to figure C will improve the figure as it brings no real information as it is. Finally, the authors should provide an excel file with the entire dataset in supplementary material (if not already done).

3/The authors should specify if the genes quantified by qRT-PCR in the parts "Mtb-AG induces early host inflammatory gene expression in the lungs" and "Mtb-AG exacerbates host inflammatory responses and promotes tissue remodeling in the lungs" have a similar trend in their RNAseq dataset. This might help to strengthen their claims and link better the different parts of the manuscript together.

4/The conditions by which the authors tested M. tuberculosis resistance against nitric oxide (NO) need improvement. The presence of sodium nitrite does not promote the generation NO unless the medium is acidified (pH5.5) (see protocol at PMID: 14671303). Unless the authors can show that their in vitro model allows the generation of a consequent amount of NO, they should repeat their experiment in acidified media or use the NO donor DETA-NO (see paper at PMID: 32482725). Alternatively, the authors could remove the data and associated comments from the manuscript.

5/Table at the end of the Supplementary Figures (after supp Fig.8). These tables lack a title to be referred to (supp Fig 9? or Table S1?). It would be appreciated if the statistical significance of the fold changes displayed could be added in the tables (or specify if they are z-score). Values for "apoptosis gene expression" are missing. It is unclear why "increased" and "decreased" values show both negative and positive numbers.

Point-by-point response to reviewers' comments

Reviewer #1

The MS don't contain any page number or line number, without page number and line number it is very difficult to guide authors to the extract query that are raised.

Response: Page numbers and line numbers are added in the revised manuscript.

General observations

Comment-1): The main theme of this MS is aggregation of mycobacteria, but how is the extent of aggregation determined, produced or maintained is not well described. When would a clump of M.tb considered as M.tb-AG should be substantiated. Is Mtb-Ag a range starting from two single M.tb to n numbers, then this range should be described and the reason for considering this range explained. There is no in vitro or in vivo control on the extent of aggregation during intracellular proliferation of M.tb. How are we to expect that M.tb-AG with n number of M.tb when proliferate would further generate more Mtb-AG and not few Mtb-SC as well. If the later case occurs then that would give mixed results. Similarly when infection is carried out with few Mtb-SC's there is no control that these M.tb-SC would not aggregate together and form M.tb-AG and thus give mixed results. Nothing has been mentioned what measures if any has been taken by authors to maintain Mtb-SC as single cells all through.

Response-1: we have elaborately described how Mtb-AG was produced, maintained, and determined the extent of aggregation in a previous publication (Ref#10). In the current/revised manuscript, we summarized this information in the Methods section (lines 325-331). Similarly, the replication kinetics of Mtb as singles and clumps in infected macrophages has been extensively demonstrated through imaging analysis of mCherry-expressing Mtb-SC and Mtb-AG (Ref#10). Since the current study extends our earlier in vitro studies, we did not include those details in this manuscript. Furthermore, it is not possible to control the aggregation of Mtb-SC or declumping of Mtb-AG after the respective inoculum is implanted into the rabbit lungs. However, the focus of this study and the most important message conveyed in the current manuscript is that the nature of Mtb (either as mostly singles or clumps) that elicits the first and earliest host response can impact the subsequent course of infection, including early host transcriptome and progression of disease pathology.

Comment-2): Heavier aerosolic droplets tend to get retarded in the pharynx and bronchi based on their size. Based on equal CFU values of Mtb-AG and M.tb-SC at time 0, the authors conclude that equal amount of M.tb was presented to the lungs for infection. If this is the case then compared to a single phagocytosis event for an n no. of Mtb containing Mtb-AG, n no of phagocytic events have to occur for M.tb-SC. This is possible only if phagocytosis is considerably slower for M.tb-AG than for M.tb-SC and if that is the case then how can pathogenesis of two differentially phagocytosed populations be compared.

Response: We are unclear about the phrase “heavier aerosol,” as mentioned by the reviewer. However, as shown by several clinical studies cited in the current manuscript, a copious amount of Mtb-aggregates is seen in clinical specimens of patients with pulmonary active, cavitary tuberculosis (Ref# 3, 4). Mtb-AG was also noticed in the cavitary wall of pulmonary granulomas (Ref#11). The differential phagocytosis of Mtb-SC and Mtb-AG and their intracellular replication kinetics within infected human macrophages has been elaborately demonstrated by our earlier publication (Ref#10). Differential phagocytosis of Mtb-SC and Mtb-AG and the subsequent response of infected phagocytes is the central theme of that publication. Our current study extends the in vitro findings by testing the hypothesis of whether Mtb-SC and Mtb-AG elicit differential pathogenesis in vivo. Results from this study suggest that the nature of Mtb in the inoculum can impact the progression of infection in the lungs.

Comment-3). A more detail description of RNAseq and network analysis done for SDEG will be helpful.

Response: We have provided more details on the RNAseq and network analysis (lines 408-434).

Comment-4). As gene expression pattern in Mtb-AG infected rabbit lungs revealed induction of both apoptosis and necrosis, so it will be better to make a percentage wise comparison between apoptosis and necrosis mediated cell death and their outcome on granuloma formation.

Response: In Figure-6, we have shown increased apoptosis, and elevated levels of necrotic cell death markers in the Mtb-AG infected lungs between 1 and 4 weeks post-infection. However, we think that correlating the percentage of apoptosis and necrosis to the outcome of granuloma formation and maturation can only be speculative. A causal association requires more extensive, additional experimental data, which is beyond the scope of this manuscript.

Comment-5). As immune cell composition has been investigated, to ascertain proper spatial localization along with increase in number of different innate immune cell types, multiparameter immunofluorescence imaging of M.tb-AG and M.tb-SC infected lung tissue sections exhibiting CD11b, CD4, F4/80, B220, M.tb, SIGLEC antibody should be carried out. This data will help in better comparison of infection exacerbated pulmonary disease pathological condition between Mtb-AG and Mtb-SC infected rabbits.

Response: In Figure-3, we have shown the percentage of key immune cells that are positive for CD11B, IBA-1(Macrophages), CD4 and CD8. Although testing additional markers would be useful, there is a limitation in the availability of antibodies for rabbits, particularly those compatible with multiparameter immunofluorescence imaging. We will perform such elaborate studies if/when compatible antibodies for rabbits are available.

Comment-6). For better analysing spatial location of different innate immune cells machine learning clustering algorithms should be used.

Response: We agree that a machine learning clustering algorithm would be beneficial to analyze images from a large number of samples. However, we neither have access to a validated algorithm to analyze the images of rabbit lung granulomas, nor it is within the scope of this manuscript to develop such an algorithm. We will perform such elaborate studies if/when validated algorithms to use in rabbit lung sections are available.

Comment-7). A complete list of used antibodies with species specificity is required in the materials and methods section. A comprehensive list of all the oligonucleotide sequences used in this study should be provided.

Response: Details of antibodies with species specificity and company are updated in the methods section lines 366-383. The list of oligonucleotide sequences used in this study is provided in Supplementary Table-7.

Comment-8). More discussion on bigger lesion size and greater induction of necrosis in Mtb-AG infected rabbits is required corresponding to macrophage, T-cell, neutrophil spatial location and their distance in between should be provided.

Response: We have revised the discussion section for better interpretation of our findings, pertinent to and supported by the published literature in the field

Comment-9). In majority of cases independent of host status differential killing of bacteria take place and majority of the granulomas are established by a single bacterium then why is lesion size of Mtb-AG infected rabbits bigger than lesion size of rabbits infected with Mtb-SC. A quantitative estimation of the lesion size and the mechanism used to measure the same should be provided.

Response: In Figure-2, we have shown the number of lung lesions, the size of lesions, and the pulmonary disease score in the rabbit lungs infected with Mtb-SC or Mtb-AG. The methodology used to obtain these data has been added in the Methods section (lines 325-331).

Comment-10). Tuberculosis spread through aerosolic droplet nuclei measuring 1-5 μm and these droplets contain 1-4 bacteria, which then reach the alveoli, are engulfed by macrophages and thereafter establish an infection leading to granuloma formation as has been shown in case of Mtb-SC. While aerosols $> 7\mu\text{m}$ tend to get retarded at the pharyngeal region which if successful in establishing an infection would in a non-macrophage cell type or in extra-pulmonary settings. Therefore what is the physiological

significance of a forced non-physiological Mtb-AG infected rabbit model, where M.tb-AG containing at least 20 M.tb forced into aerosol formation for infection. Such M.tb-AG under normal condition cannot be spread as aerosol droplet from one person to another.

Response: We have a different perspective on the infectious nature of Mtb clumps, which is supported by several clinical studies that are also cited in the current manuscript (Ref#3, 4 and 11). In most of the experimental studies, single-cell suspensions of Mtb were generated to facilitate a more straightforward quantification of the bacilli used for infection and microbial characterization. Therefore, the practice in the field has been to use Mtb grown, to the extent possible, as single cells. Consequently, the potential role of Mtb aggregation in driving the outcome of host-pathogen interactions and disease pathogenesis has essentially been overlooked and understudied. A copious amount of Mtb-aggregates is seen in the clinical specimen of patients with pulmonary active, cavitary tuberculosis (Ref# 3, 4) and the cavitary wall of pulmonary granulomas (Ref#11). In fact, clinical studies show that patients with active pulmonary TB produced aerosols with ≥ 10 Mtb CFU consistently had higher sputum AFB smear grades and transmit more efficiently than those making aerosols with less than 10 CFU (Ref#3). Therefore, the nature of bacteria in the infectious inoculum clearly affects the outcome of the host response to Mtb, and bacterial aggregation leads to a more severe disease. Thus, our experimental study is highly relevant to the differential outcome of infection by Mtb as singles or clumps in a relevant-to-human animal model. Findings from this study suggest that the nature of Mtb in the inoculum can impact the progression of infection in the lungs.

Comment-11). How is the M.tb-SC maintained as single cells once it enters inside the respiratory track of the rabbit?

Response: In this study, we neither claimed that Mtb-SC was maintained at its single-cell status after entering into the rabbit lungs, nor it is possible to keep Mtb-SC as purely single bacteria in vivo after infection.

Comment-12). Role of TNF alpha cannot be understated in case of initial disease spread. Why TNF alpha has been measured after 4 wk of infection and not after 1 wk of infection when bacteria starts to get disseminated.

Response: Our intention to measure TNFa was to correlate the expression of this proinflammatory cytokine with the granulomatous response. At 4 weeks post-infection, the lung granulomas show distinct morphometric features between Mtb-SC and Mtb-AG infected rabbits. However, we do not have any evidence that Mtb is disseminated at 1 week post-infection in our rabbit model of pulmonary infection.

Specific points

Comment-1. Introduction-last para-3rd line-'aerosolized Mtb-AG': If Mtb-SC is single

M.tb then what is M.tb-AG two bacteria onwards. This is unclear. Proper experiments should be included which makes a clear distinction when would a M.tb clump be designated as M.tb-AG.

Response: We elaborately presented the enumeration of bacilli in Mtb-AG in a previous publication (Ref#10), which is also mentioned the methods (lines 325-331)

Comment-2. Introduction-last para-4th line-'similar number of M.tb-SC: what is meant by similar number of M.tb-SC. If M.tb-AG constitutes a specific number of M.tb forming an aggregate then this can be considered, but if M.tb-AG is a range of M.tb aggregates then how are the authors monitoring the similar number of M.tb-SC each time. On the contrary if M.tb-AG is a fixed number of M.tb containing aggregate then how do the authors obtain this specific aggregate should be stated explicitly. If M.tb-SC contains same number of M.tb singly that are present in M.tb-AG then how do the authors make sure that these M.tb-SC's do not form mini M.tb-AG's within themselves. Besides how do the authors also control the formation of macro M.tb-AG by coming together of 2 or more M.tb-AG.

Response: We elaborately presented the enumeration of bacilli in Mtb-AG in a previous publication (Ref#10). The total number of bacteria was adjusted to be similar, although they were used as two different morphological units (i.e, singles or aggregates) in experiments. Therefore, the initial inoculum prepared for in vitro (Ref#10) or in vivo (this study) infection had a similar total number of live bacteria.

Comment-3. Results-Section 1-3rd line- if the CFU is similar then the no. of M.tb whether it be M.tb-AG or M.tb-SC are similar. Arbitrarily considering that M.tb-AG is constituted by 5 M.tb, then CFU would be similar if the extent of phagocytosis in M.tb-AG is 5 fold less than M.tb-SC. If the phagocytosis is similar then the CFU would be 25 times than that of M.tb-SC. Besides to obtain CFU, M.tb from within the macrophage is obtained by lysing and then plated. So the CFU would be similar only when the extent of phagocytosis of M.tb-AG would be that many fold less as to the number of M.tb present in the M.tb-AG. If the M.tb-AG is a range then it would be more complex proposition. If one colony is formed by each M.tb-AG then the colony size would vary according to the number of M.tb present in each M.tb-AG. In such a case it would not be a true CFU.

Response: We elaborately presented the enumeration of bacilli in Mtb-AG in a previous publication (Ref#10). The total number of bacteria was adjusted to be similar, although they were used as two different morphological units (i.e, singles or aggregates) in experiments. Therefore, the initial inoculum prepared for in vitro (Ref#10) or in vivo (this study) infection had a similar total number of live bacteria as determined by CFU assay.

Comment-4. Results-Section 1-4th line- The significant increase in cfu could be attributed to lower doubling time of Mtb-AG as compared to M.tb-SC as well as

comparable number of M.tb being present inside macrophages at 0 timepoint. If the latter is true then phagocytosis of M.tb-AG is much less than M.tb-SC at a given time. Therefore, raising the question of feasibility of comparing intracellular proliferation when the extent of uptake is different.

Response: The differential phagocytosis of Mtb-SC and Mtb-AG and their intracellular replication kinetics within infected human macrophages has been elaborately demonstrated by our earlier publication (Ref#10). Differential phagocytosis of Mtb-SC and Mtb-AG and the subsequent response of infected phagocytes is the central theme of that publication. Our current study extends the in vitro findings by testing the hypothesis of whether Mtb-SC and Mtb-AG elicit differential pathogenesis in vivo. Results from this study suggest that the nature of Mtb in the inoculum can impact the progression of infection in the lungs. Although the increase in CFU could be attributed to the difference in the doubling time between Mtb-SC and Mtb-AG, it requires additional, more detailed molecular analysis to confirm such a notion, which will be pursued in future research.

Comment-5. Results-Section 1-8th line-‘imaging fluorescent’-representative images should be provided in the supplementary or main text.

Response: As requested by this reviewer, we have provided representative images of Mtb-SC and Mtb-AG in infected rabbit lungs (Supplementary Figure-1A-D).

Comment-6. Results-Section 1-8th line-‘net bacillary load’-how do the authors ensure that the M.tb-AG is taken up by macrophages and not by neutrophils. In one of their previous studies, it was shown that M.tb-AG is taken up by cells isolated from donor blood using CD14+ MACS beads which they termed MOM. It is to be noted that neutrophils do express CD14 to varying extents and would thus be isolated along with MOM.

Response: In this manuscript, we did not analyze the differential uptake of Mtb by macrophages or neutrophils, nor did we exclude the possibility that neutrophils can phagocytose Mtb. Instead, we provided the net bacillary load and related disease pathology in the lungs of rabbits infected with Mtb-SC or Mtb-AG at various time points.

Comment-7. Results-Section 2-5th line-‘polymorphonuclear cell accumulation’-using a distinct set of markers the authors need to show conclusively that the uptake of non-physiological Mtb-AG occurs by macrophages and not by any other cells type such as neutrophils.

Response: We do not understand what is “ non-physiological Mtb-AG”. However, as shown in Figure 3D, our histologic examination of H&E-stained lung sections revealed the presence of multiple, coalescent lesions with several prominent necrotic foci with abundant polymorphonuclear cell accumulation, preferentially in the Mtb-AG-infected

animals. This does not imply any concept related to phagocytosis of Mtb by neutrophils or macrophages.

Comment-8. Results-Section 2-8th line-‘small clumps’- why would these clumps not be designated under M.tb-AG and thereby whatever pathophysiology has been attributed to M.tb-AG from initial stage should be attributed to M.tb-SC from the second granuloma onwards.

Response: We do not understand the significance of this question and the definition of “second granuloma onwards” mean. However, the “small clumps” identified in histological analysis were qualitative rather than quantitative to describe how the bacillary population was distributed in the rabbit lungs infected with Mtb-SC versus Mtb-AG.

Comment-9. Results-Section 2-10th line-‘subpleural...Mtb-SC’-in the corresponding figures the authors have only provided H-E staining and provided pointers for macrophages and neutrophils. The authors need to provide conclusive immunohistochemistry and/or immunofluorescence data exhibiting the macrophage to neutrophil distribution and the extent of infection therein for such sections from Mtb-AG and M.tb-SC infected lung sections.

Response: Macrophages and polymorphonuclear cells can be distinguished by H&E stained histological imaging based on peculiar cell structures and nuclear morphology. This method has been routinely used in many clinical pathology research. We have provided the differential distribution of various immune cells, including macrophages and T cell subsets, in Figure-4.

Comment-10. Results-Section 2-13th line-‘more...infection’- M.tb-AG exhibits more extensive cell recruitment as compared to Mtb-SC. M.tb infection is a form of stress. Highly stressed neutrophils are known to secrete myeloperoxidases (MPO) that then can induce severe inflammation and extensive cell recruitment. Authors should provide experiments to negate such a proposition.

Response: We appreciate the reviewers’ proposition on the role of Mtb infection in generating “highly stressed” neutrophils and their role in inflammation. However, that is not within the scope of this manuscript. However, our future studies would address the mechanistic role of MPO and other neutrophil effector molecules on lung inflammation during Mtb infection.

Comment-11. Results-Section 2-16th line-‘flow cytometry...immunohisto’-the authors used single cell suspensions to analyse the different cell types in the infected lungs. A flow cytometry data corresponding to stained M.tb along with other cell type is required to establish the context of infection here.

Response: As we clearly mentioned, the purpose of flow cytometry of single-cell suspension and immunohistochemistry in Figures 4A and C was to determine the immune cell distribution in Mtb-AG or Mtb-SC infected rabbit lungs. The lung bacillary loads in these rabbits are provided in Figure-1.

Comment-12. Results-Section 2-20th line-‘significantly....four weeks’-naive neutrophils are also known to express higher levels of CD11b compared to the activated/stressed ones. Hence in 4weeks for M.tb-SC when true granuloma is formed lots of naive neutrophils are recruited which may not be the case for M.tb-AG which already could be inside a activated/stressed neutrophil and therefore does not induce further recruitment of naive neutrophils. Such propositions drawn from the provided data should be experimentally negated.

Response: We do not understand the significance of this question since this “negation” is out of scope for the current manuscript. We did not propose any such mechanisms nor disagree that CD11b is expressed in neutrophils.

Comment-13. Results-Section 3-8th line-‘inflammatory....upregulated’-since RNA seq analysis was carried out from different cell types in the lung tissue it would be hard to envisage whether the enumerated expression changes in Mtb-AG infected lungs as compared to Mtb-SC infected lungs is from the same cell type or not. So experiments exhibiting that for the presented data, both M.tb-AG and M.tb-SC is taken up by same cell type should be included.

Response: The purpose of RNAseq data was to determine the genome-wide transcriptional changes associated with early stages of Mtb-SC or Mtb-AG infection in rabbit lungs. We agree that single-cell RNAseq analysis of Mtb-SC versus Mtb-AG infected immune cells is helpful to determine the effect of infection on such cells. However, that was not the focus of this manuscript. Future studies would address these mechanistic aspects.

Comment-14. Results-Section 3-12th line-‘induction....necrosis’-In context of Mtb-AG infection only the infected cells, die at the initial stage. So how is it that the same cell is dying by necrosis as well as apoptosis simultaneously.

Response: We did not mention that the same cell is dying of apoptosis and necrosis; instead, we report the presence of both apoptosis and necrosis in Mtb-AG infected rabbit lungs. Investigating the bacterial factors that contribute to various modes of host cell death was not the focus of this manuscript. Future studies would address these mechanistic aspects.

Comment-15. Results-Section 3-17th line-‘macropinocytosis signalling’-From figure 1 it is evident that M.tb-AG is taken up through reduced phagocytosis. This could be

circumvented if it is to be considered that activated macrophages which are Iba+ engulf these M.tb-AG. It is known that activated macrophages/monocytes uptake mycobacteria through macropinocytosis as the receptor mediated uptake for these clumps may not be possible and it also allows for uptake to a large extent. So it is surprising that genes corresponding to macropinocytosis is downregulated in M.tb-AG infected lungs.

Response: The purpose of RNAseq data was to determine the genome-wide transcriptional changes associated with early stages of Mtb-SC or Mtb-AG infection in rabbit lungs. We do not fully understand the role of micropinocytosis in the pathogenesis of Mtb-AG infected rabbit lungs. Future studies would address these mechanistic aspects.

Comment-16. Results-Section 3-19th line-'PPAR'-how do the authors specifically say that this upregulation of PPAR pathway genes are from Mtb-AG infected macrophage/monocytes and not from other cell types such as neutrophils as similar upregulation is observed there as well when they are dying by apoptosis.

Response: As we mentioned in lines 170-172, the genes involved in PTEN, PPAR, calcium, and CD27 signaling in immune cells were upregulated in Mtb-AG infected lungs. We did not make any argument that PPAR is expressed specifically from macrophage/monocytes.

Comment-17. Results-Section 4-3rd line-'upregulated.... inflammatory'-through the data presented in Table 2, the authors state that genes responsible for Th1 response are downregulated in Mtb-AG infected lungs. It is known that Th1 response corresponds to a pro-inflammatory response. But here the authors now show that genes corresponding to inflammatory response is upregulated in the lungs upon Mtb-AG infection. What would be the explanation to this contradiction.

Response: While Th1 is a pro-inflammatory response associated with host protection during Mtb infection, exacerbated inflammation mediated by molecules such as CRP, S100A8, and others that we mentioned in this manuscript can dampen this host protective response. Various host-destructive processes, such as necrosis of infected cells, can exacerbate inflammation and promote disease pathology in the lungs while dampening the effect of Th1 mediated protective response. The causal role of pathways contributed by Th1 and inflammatory molecules to the pathogenesis of Mtb in rabbit lungs warrants further studies.

Comment-18. Results-Section 4-9th line-'while....lungs'- this is strong statement to make based on just few gene upregulation data that too from the gross lungs and not the niche cells. This statement should be altered.

Response: As we clearly mentioned, our observations suggest that Mtb-AG induces a more robust host inflammatory response, while the antimicrobial response genes were

preferentially upregulated in the Mtb-SC infected lungs. We did not specifically correlate the findings to a specific immune cell population nor conclude that these are the exclusive list of genes involved in those host responses to Mtb infection.

Comment-19. Results-Section 5-4th line-‘percentage...Mtb-AG’-Previously the authors state that genes corresponding to both apoptosis and necrosis were upregulated in the lungs upon infection with M.tb-AG. But here they exhibit only biochemical markers corresponding to apoptosis, this is surprising. Besides both Mtb-AG and Mtb-SC have been shown to form granulomas wherein granulomas are known to have a necrotising centre. Through these data do the authors suggest that granulomas of M.tb-AG have a apoptotic and dead cell core instead of a necrotic core which they had suggested earlier.

Response: The RNAseq data presented was from rabbit lungs at 24hours post-infection. However, the data on host cell death markers presented in Figure 6 is from rabbit lungs at 1 or 2 and 4 weeks post-infection. As shown in Figure-3, the histology analysis showed necrotic granulomas in both Mtb-SC and Mtb-AG infected rabbit lungs, although an elevated necrosis inflammatory response was noted in the latter case. It is possible that the necrotic core of the granulomas was comprised of cell debris generated by apoptosis and necrosis. However, the relative contribution of these two modes of cell death on granuloma formation remains unknown, and additional/detailed studies are needed to confirm this aspect.

Comment-20. Results-Section 5-7th line-‘nitric oxide...infection’-upon phagocytosis of a pathogen the prime macrophage response is to kill the bug by barging it with ROS and RNI. Overproduction of ROS and RNI in a cell can lead to apoptosis of the cell. The precursor for RNI is NO. Considering that upon Mtb-AG infection the infected cells are the ones that die due to apoptosis it is quite surprising to find that these cells although dying by apoptosis exhibits reduced levels of NO. Additionally authors in earlier figure have shown Iba1+ cells in the lungs of M.tb-AG infected cells which is indicative of activated macrophages and monocytes. It is known that such activated macrophages and monocytes have higher RNI and ROS.

Response: Although ROS and RNS produced by activated macrophages exert antimicrobial response during Mtb infection, it was not known how that response would be impacted if/when the Mtb is presented as aggregates. As we have described, conventionally, researchers make single-cells of Mtb by growing the bacteria in the presence of detergents or mechanically disrupting the bacterial clumps. Thus, most of the published data on host-Mtb interactions were derived from experiments conducted with mostly single bacteria. In contrast, we report a differential host response upon infection with Mtb presented as aggregates. However, the mechanistic association between clumping phenotype and alteration of specific cell function such as ROS and RNS production by macrophages warrants more detailed future research.

Comment-21. Results-Section 6-21th line-'four weeks'-Granuloma formation occurs both in Mtb-AG and Mtb-SC infected lungs. Overexpression of MMP's are associated with granulomas. Therefore the higher levels of MMP expression in Mtb-AG infected lungs 4weeks post infection is interesting.

Response: We agree with this comment. Increased MMP expression in Mtb-AG infected rabbit lungs could be associated with more tissue destruction and larger granulomas than Mtb-SC infected animals. However, more studies are needed to validate this hypothesis mechanistically.

Reviewer #2

Comment-1/ As one clump gives one CFU, more bacteria could have been present in the Mtb-AG in comparison to Mtb-SC and being responsible to the phenotypes observed. But the authors have made the effort to show the good correlation between CFUs, CEQ and fluorescence to convince us that the number of bacteria compared is the same.

Response: We appreciate and thank the reviewer for understanding the intricacies of this type of work reported in this manuscript.

Comment-2/ Figure 4. The figure and figure legends should be modified to improve clarity. It is unclear what do "increased" and "decreased" values mean and why those categories show both negative and positive values. Also, it is not clear to me if the data presented are fold changes or z-scores. In the legend, reference is made to Figure 4D but there is not figure 4D. The addition of gene names to figure C will improve the figure as it brings no real information as it is. Finally, the authors should provide an excel file with the entire dataset in supplementary material (if not already done).

Response: We agree with the reviewer's comment. The biological function, such as cell viability, is regulated by a group of associated genes. Within this group of genes, some are upregulated, and others are downregulated. However, the cumulative effect of all such up and down-regulated genes determines the directionality of the biological function (i.e., increased or decreased). Therefore, the increased biological function does not necessarily translate into upregulation of all genes and vice versa. Downregulation of some genes (e.g, negative regulators) can actually increase/upregulate a biological function. However, in the context of this manuscript, to interpret our data clearly and understandable, we added the information of the original Figure-4 in Table-1 and removed the original figure 4. As suggested by the reviewer, we have provided excel files for all the network/pathway dataset as Supplementary Figures 1-7.

Comment-3/The authors should specify if the genes quantified by qRT-PCR in the parts “Mtb-AG induces early host inflammatory gene expression in the lungs” and “Mtb-AG exacerbates host inflammatory responses and promotes tissue remodeling in the lungs” have a similar trend in their RNAseq dataset. This might help to strengthen their claims and link better the different parts of the manuscript together.

Response: We appreciate this comment. As suggested, we checked and noticed that the directionality of expression of several of these genes is consistent between the RNAseq and the qPCR data. This information is added in lines 185-186 of the revised manuscript.

Comment-4/The conditions by which the authors tested *M. tuberculosis* resistance against nitric oxide (NO) need improvement. The presence of sodium nitrite does not promote the generation NO unless the medium is acidified (pH5.5) (see protocol at PMID: 14671303). Unless the authors can show that their in vitro model allows the generation of a consequent amount of NO, they should repeat their experiment in acidified media or use the NO donor DETA-NO (see paper at PMID: 32482725). Alternatively, the authors could remove the data and associated comments from the manuscript.

Response: As suggested by the reviewer, we have removed the data and associated comments related to nitric oxide exposure of *Mtb* in the revised manuscript.

Comment-5/Table at the end of the Supplementary Figures (after supp Fig.8). These tables lack a title to be referred to (supp Fig 9? or Table S1?). It would be appreciated if the statistical significance of the fold changes displayed could be added in the tables (or specify if they are z-score). Values for “apoptosis gene expression” are missing. It is unclear why “increased” and “decreased” values show both negative and positive numbers.

Response: We have revised the supplementary tables to include captions and other details.

REVIEWERS' COMMENTS:

Reviewer #1 (Remarks to the Author):

Comment to Response 4

The authors mostly refer to their previous *in vitro* work for most of their standardizations. They should have done the standardization with respect to apoptotic or necrotic death in that work itself and if not asked for there they should have done it here and included in supplementary. The statement that the cells could die by apoptosis or necrosis is unacceptable. Indeed, the extent of apoptosis/necrosis in context of granuloma is important because for Mtb elimination within granuloma necroptosis and necrosis is crucial. While apoptosis is crucial with respect to Mtb proliferation within the cells. Besides in a context where it is unclear whether the infected cells are neutrophils or macrophages this becomes more crucial.

Comment to Response 6

Lack of antibodies cannot be an excuse for not carrying out analysis of more than one cell type specific markers, since it would only be through such study that the authors can conclusively state which cell type in actuality is infected. Additionally, it is hard to believe that F4/80, CD4, SIGLEC antibodies for rabbit is unavailable.

Comment to Response 10

In their earlier work they had carried out lot of Matlab programming and mathematical modelling work. So being an *in vivo* continuation to that work such clustering work should not be out of scope for the authors, unless they feel that it's not required for a Comm Biol publication. A practice in the field need not be departed from if it is non-physiological in context of phagocytosis of mycobacteria by macrophages. Literature shows that mycobacterial infection of single cells has a greater chance of successful pathogenesis as opposed to clumps. Additionally clumped mycobacteria are generally taken up by neutrophils. So, in an *in vivo* context even if it is considered that upon proliferation of Mtb some amount of aggregated Mtb can occur then if they are taken up by neutrophils, while single cells are taken up by macrophages, we cannot compare the two under any circumstance.

Comment to Response 11

If Mtb-SC cannot be kept as single after infection and Mtb-AG can also generate single AG dislodged SC, then we would always get a mixed response so it would be hard to make an *in vivo* demarcation for such mixed results without further Matlab algorithms and allied which the authors are reluctant to do.

Comment to Specific response 2

If quantitated fluorescence was used to take equal amount of Mtb for both AG and SC then also the uptake of Mtb-AG would occur in one event while uptake of equal number of Mtb-SC would take that many number of events. So it is unlikely that phagocytosis (if at all) of an Mtb-AG having n number of Mtb and n no. of phagocytosis events for n number of Mtb-SC would take similar time. In the presented context how can the CFU then be comparable unless the phagocytosis event of M.tb-AG is n times slower.

Comment to Specific response 6

It is to be noted that the authors acknowledge that the uptake of Mtb-AG could be into neutrophils as well. Besides the authors also acknowledge that Mtb-SC can form aggregates *in vivo* and Mtb-AG can dislodge to generate few Mtb-SC. Besides whether the uptake of Mtb-AG is phagocytosis or macropinocytosis is also not ascertained. In such a scenario of so many ifs and buts the physiological significance of infection by an *in vitro* generated Mtb-AG is irrelevant in context of phagocytosis.

Comment to Specific response 9

The point in question was neutrophils not T cells. No conclusive evidence have been provided by the authors to nullify that Mtb-AG is taken up by macrophages and not neutrophils. Based on the authors

submission that PMN cells in general can take up Mtb-AG whereas Mtb-SC is strictly taken up by macrophages, the study becomes irrelevant in an in vivo context of pathogenesis.

Comments to Specific response 10

While at one hand authors cannot prove that neutrophils are not involved in the uptake of Mtb-AG and on the other hand they admit that PMN's can take up Mtb-AG which includes neutrophils the raised point cannot be put out of context and kept unaddressed.

Comment to specific response 13

The RNAseq data corresponding to infected cells is very important rather than transcriptional changes in the lung cells in general. This is because the authors are comparing a non-physiological Mtb-AG uptake by any form of PMN to uptake of Mtb-SC by macrophages.

Comment to specific response 15

Definitive conclusions cannot be drawn from inconclusive data and conclusive experiments stated to be out of context.

Comments to specific response 16

Since in the authors admit that Mtb-AG could be taken up by any PMN including neutrophils, the manuscript should be modified such that the essence of increased pathogenicity of Mtb-AG originates from its predominant uptake by the neutrophils.

Comments to specific response 17

How would Mtb-AG at one end upregulate a Th1 response which is known to eliminate pathogen, while at the same time exhibit increased infectivity and pathogenicity.

Comment to specific response 18

Predominantly most literature describe infection by Mtb-SC wherein mycobacteria can subvert the immune response and dampen anti-microbial response thus establishing an infection. But the authors claim that Mtb-SC exhibits upregulation of anti-microbial genes in Mtb-SC infected macrophages.

Response to reviewer comments

Reviewer #1

1). Reviewer Comment to Response 4:

The authors mostly refer to their previous *in vitro* work for most of their standardizations. They should have done the standardization with respect to apoptotic or necrotic death in that work itself and if not asked for there they should have done it here and included in supplementary. The statement that the cells could die by apoptosis or necrosis is unacceptable. Indeed, the extent of apoptosis/necrosis in context of granuloma is important because for Mtb elimination within granuloma necroptosis and necrosis is crucial. While apoptosis is crucial with respect to Mtb proliferation within the cells. Besides in a context where it is unclear whether the infected cells are neutrophils or macrophages this becomes more crucial.

Response: Standardization of necrotic and apoptotic cell death has been reported in our previous study (Ref#10). The extent of host cell death by apoptosis and by necrosis are provided in Figure-6, which suggest that both apoptosis and non-apoptotic cell death markers, such as LDH release and cytotoxicity, were significantly elevated in Mtb-AG infected lungs. Previous studies have shown that virulent Mtb can promote infected host cell lysis by apoptosis, necrosis and necroptosis (Ref# 23, 24). This is consistent with our reports in this study.

2). Reviewer Comment to Response 6:

Lack of antibodies cannot be an excuse for not carrying out analysis of more than one cell type specific markers, since it would only be through such study that the authors can conclusively state which cell type in actuality is infected. Additionally, it is hard to believe that F4/80, CD4, SIGLEC antibodies for rabbit is unavailable.

Response: We have already provided the data for the frequency of immune cells such as Iba1+ (macrophages), CD11B+ (monocytes/macrophages), CD4+ and CD8+ (T-cells) cells in Mtb-SC and Mtb-clump infected rabbit lungs (Figure-4).

3). Reviewer comment to Response 10

In their earlier work they had carried out lot of Matlab programming and mathematical modelling work. So being an *in vivo* continuation to that work such clustering work should not be out of scope for the authors, unless they feel that it's not required for a Comm Biol publication. A practice in the field need not be departed from if it is non-physiological in context of phagocytosis of mycobacteria by macrophages. Literature shows that mycobacterial infection of single cells has a greater chance of successful pathogenesis as opposed to clumps. Additionally, clumped mycobacteria are generally taken up by neutrophils. So, in an *in vivo* context even if it is considered that upon proliferation of Mtb some amount of aggregated Mtb can occur then if they are taken up by neutrophils, while single cells are taken up by macrophages, we cannot compare the two under any circumstance.

Response: We have not evaluated the differential phagocytosis of Mtb-AG and Mtb-SC by macrophages versus neutrophils in this study. As mentioned in our earlier response to a relevant query, several clinical studies have reported the presence of Mtb as aggregates, which promoted more bacillary transmission and host cell death. We have discussed this aspect elaborately in the manuscript. However, we disagree that "mycobacterial infection of single cells has a greater chance of successful pathogenesis as opposed to clumps". Further, we did not claim that "clumped mycobacteria are generally taken up by neutrophils" and thus, it is irrelevant to discuss this aspect in the manuscript. To clarify this point, we have added the following sentence: "Phagocytes, such as macrophages, dendritic cells (DC) and neutrophils may utilize various phagocytic mechanisms to engulf Mtb^{22,23}. For example, macrophages employ different

mode of phagocytosis through Fc and C3 receptors, as reported earlier^{23,24}. Importantly, the fate of intracellular Mtb is impacted by the type of phagocytic receptor involved in the bacterial uptake^{23,25}. However, the relative contribution of various phagocytes and their cell surface receptors involved in the uptake of Mtb-AG versus Mtb-SC is yet to be determined” (page-14).

4). Reviewer comment to Response 11

If Mtb-SC cannot be kept as single after infection and Mtb-AG can also generate single AG dislodged SC, then we would always get a mixed response so it would be hard to make an in vivo demarcation for such mixed results without further Matlab algorithms and allied which the authors are reluctant to do.

Response: Our central hypothesis is that the nature of Mtb at the time of its first encounter of the host cells (i.e early host-pathogen interactions) determine the later outcome following infection. Thus, uptake of Mtb as single cell versus aggregates may have differential outcome in the infected host. In a previous study (Ref#10), we demonstrated that uptake of a single large Mtb-AG was more cytotoxic than pickup of similar number of Mtb as single cells. Consistent with those in vitro findings, we observed elevated host inflammatory response and disease pathogenesis in rabbit lungs infected with Mtb-AG, compared to Mtb-SC infection.

Specific points

1). Comment to specific response 2

If quantitated fluorescence was used to take equal amount of Mtb for both AG and SC then also the uptake of Mtb-AG would occur in one event while uptake of equal number of Mtb-SC would take that many number of events. So it is unlikely that phagocytosis (if at all) of an Mtb-AG having n number of Mtb and n no. of phagocytosis events for n number of Mtb-SC would take similar time. In the presented context how can the CFU then be comparable unless the phagocytosis event of M.tb-AG is n times slower.

Response: We have not determined the phagocytosis event of Mtb-AG in this study. However, in a previous in vitro study (Ref# 10), using time-lapsed microscopic imaging, we have clearly demonstrated that the aggregation state of Mtb determines the fate of infected human macrophages and that Mtb aggregates can grow rapidly inside dead and infected macrophages.

2). Comment to specific response 6

It is to be noted that the authors acknowledge that the uptake of Mtb-AG could be into neutrophils as well. Besides the authors also acknowledge that Mtb-SC can form aggregates in vivo and Mtb-AG can dislodge to generate few Mtb-SC. Besides whether the uptake of Mtb-AG is phagocytosis or macropinocytosis is also not ascertained. In such a scenario of so many ifs and buts the physiological significance of infection by an in vitro generated Mtb-AG is irrelevant in context of phagocytosis.

Response: We believe that the nature of Mtb (i.e AG versus SC) impacts phagocyte response and is relevant to the pathogenesis of TB. Our view is supported by several clinical studies, which are cited in the current manuscript (Ref#3, 4, 10 and 11). In most of the laboratory experimental studies, single-cell suspensions of Mtb were generated to facilitate a more straightforward and comfortable way of bacterial quantification. Thus, the practice in the field has been to use Mtb grown, to the extent possible, as single cells in media containing detergents. Consequently, the potential role of Mtb-AG in driving the outcome of host-pathogen interactions and disease pathogenesis has essentially been overlooked and understudied. A copious amount of Mtb-aggregates is seen in the clinical specimen of patients with pulmonary active, cavitary tuberculosis (Ref# 3, 4) and the cavitary wall of pulmonary granulomas

(Ref#11). In fact, clinical studies show that patients with active pulmonary TB produced aerosols with ≥ 10 Mtb CFU consistently had higher sputum AFB smear grades and transmit more efficiently than those making aerosols with less than 10 CFU (Ref#3). Therefore, the nature of bacteria in the infectious inoculum clearly affects the outcome of the host response to Mtb, and bacterial aggregation leads to a more severe disease. In the present study, we investigated the overall pathogenicity of Mtb singles and clumps instead of focusing on single cell type. We reported that the Mtb clumps subvert host immune response toward more inflammatory cause more severe infection as compared to Mtb-singles. Thus, our experimental study is highly relevant to the differential outcome of infection by Mtb as singles or clumps in a relevant-to-human animal model.

3). Comment to specific response 9

The point in question was neutrophils not T cells. No conclusive evidence have been provided by the authors to nullify that Mtb-AG is taken up by macrophages and not neutrophils. Based on the authors submission that PMN cells in general can take up Mtb-AG whereas Mtb-SC is strictly taken up by macrophages, the study becomes irrelevant in an in vivo context of pathogenesis.

Response: We did not mention in the manuscript that “PMN cells in general can take up Mtb-AG whereas Mtb-SC is strictly taken up by macrophages”, rather, our data suggest that the nature of bacteria in the infectious inoculum clearly affects the outcome of the host response to Mtb, and bacterial aggregation leads to a more severe disease. In the present study, we investigated the overall pathogenicity of Mtb singles and clumps instead of focusing on the phagocytosis of a specific cell type. Our experimental study, conducted elaborately in a relevant-to-human animal model, is highly relevant to understand the differential pathogenesis elicited by Mb-AG versus Mtb-SC.

4). Comment to specific response 10

While at one hand authors cannot prove that neutrophils are not involved in the uptake of Mtb-AG and on the other hand they admit that PMN's can take up Mtb-AG which includes neutrophils the raised point cannot be put out of context and kept unaddressed.

Response: The focus of this article was not to prove whether neutrophils are uniquely involved in the uptake of Mtb-AG or Mtb-SC. Rather, we tested the hypothesis that the nature of Mtb (AG versus SC) in the infectious inoculum can impact the outcome of infection. Results of this study suggest that Mtb-AG produce more severe inflammatory response and tissue destruction in the lungs of infected animals, compared to Mtb-SC infection. The mechanical input of individual phagocytes, including neutrophils, will be evaluated in subsequent studies.

5). Comment to specific response 13

The RNAseq data corresponding to infected cells is very important rather than transcriptional changes in the lung cells in general. This is because the authors are comparing a non-physiological Mtb-AG uptake by any form of PMN to uptake of Mtb-SC by macrophages.

Response: It was a misconception that we were comparing “non-physiological Mtb-AG uptake by PMN to uptake of Mtb-SC by macrophages”. Firstly, Mtb-AG is physiological as evidenced by several clinical and epidemiological tuberculosis on human subjects. Secondly, it remains unknown whether PMN and macrophages preferentially uptake Mtb-AG and Mtb-SC, respectively. The purpose of RNAseq is to determine the very early (24hours post inoculation of bacteria) genome-wide transcriptional changes in the lungs of Mtb-SC versus Mtb-AG infected rabbits, compared to uninfected controls, as mentioned in the manuscript. The mechanistic role of individual phagocyte types, including neutrophils, on the differential pathogenesis upon

infection by Mtb-SC versus Mtb-AG will be evaluated in a future study.

6). Comment to specific response 15

Definitive conclusions cannot be drawn from inconclusive data and conclusive experiments stated to be out of context.

Response: In this article, we tested the hypothesis that the nature of Mtb (AG versus SC) in the infectious inoculum can impact the outcome of infection. Results of this study suggest that Mtb-AG produce more severe inflammatory response and tissue destruction in the lungs of infected animals, compared to Mtb-SC infection.

7). Comment to specific response 16

Since in the authors admit that Mtb-AG could be taken up by any PMN including neutrophils, the manuscript should be modified such that the essence of increased pathogenicity of Mtb-AG. originates from its predominant uptake by the neutrophils.

Response: The focus of this article was not to prove whether neutrophils are uniquely involved in the uptake of Mtb-AG. However, as a key innate immune cell, neutrophils are involved in the phagocytosis of infectious agents, including Mtb. Our experimental data suggest differential pathogenesis following infection with Mtb-SC and Mtb-AG in rabbit lungs. However, the mechanistic role of individual phagocyte types, including neutrophils warrant detailed studies, which will be conducted in the future.

8). Comment to specific response 17

How would Mtb-AG at one end upregulate a Th1 response which is known to eliminate pathogen, while at the same time exhibit increased infectivity and pathogenicity.

Response: It is well known that excessive production of Th1 type molecules leads to inflammation during infectious and non-infectious diseases. While Th1 is a pro-inflammatory response associated with host protection during Mtb infection, exacerbated inflammation mediated by molecules such as CRP, S100A8, S100A12, IL17A and others that we mentioned in this manuscript can dampen this host protective response. Various host-destructive processes, such as necrosis of infected cells, can exacerbate inflammation and promote disease pathology in the lungs while dampening the effect of Th1 mediated protective response as shown in Table-2. Thus, optimal regulation of Th1 response is crucial in determining the host-protective versus host-destructive effects of this response. The causal role of specific Th1-molecule mediated pathways and inflammatory molecules on the differential pathogenesis of Mtb-AG versus Mtb-SC in rabbit lungs warrants further studies.

9). Comment to specific response 18

Predominantly most literature describe infection by Mtb-SC wherein mycobacteria can subvert the immune response and dampen anti-microbial response thus establishing an infection. But the authors claim that Mtb-SC exhibits upregulation of anti-microbial genes in Mtb-SC infected macrophages.

Response: In a previous study (Ref#10), we showed that Mtb-AG triggered more macrophage necrosis and associated inflammatory response, compared to Mtb-SC. The RNAseq data in the present study suggest that one way the host controls Mtb-SC is by upregulating antimicrobial response in the immune cells. Consistently, we observed less bacterial load and disease pathology in rabbit lungs infected with Mtb-SC, compared to Mtb-AG. We did not correlate the RNAseq findings to a specific immune cell population (e.g macrophages) nor conclude that these are the exclusive list of genes involved in those host responses to Mtb infection.